# Cross-species analysis of viral nucleic acid interacting proteins identifies TAOKs as innate immune regulators

Friederike L. Pennemann[1], Assel Mussabekova[2], Christian Urban [1], Alexey Stukalov [1], Line Lykke Andersen[1], Vincent Grass [1], Teresa Maria Lavacca [1], Cathleen Holze[3], Lila Oubraham [1], Yasmine Benamrouche[2], Enrico Girardi [4], Rasha E. Boulos [5], Rune Hartmann [6], Giulio Superti-Furga [4,7], Matthias Habjan[3], Jean-Luc Imler [2], Carine Meignin [2] & Andreas Pichlmair [1,3,8 ✉]

The cell intrinsic antiviral response of multicellular organisms developed over millions of years and critically relies on the ability to sense and eliminate viral nucleic acids. Here we use an affinity proteomics approach in evolutionary distant species (human, mouse and fly) to identify proteins that are conserved in their ability to associate with diverse viral nucleic acids. This approach shows a core of orthologous proteins targeting viral genetic material and species-specific interactions. Functional characterization of the influence of 181 candidates on replication of 6 distinct viruses in human cells and flies identifies 128 nucleic acid binding proteins with an impact on virus growth. We identify the family of TAO kinases (TAOK1, −2 and −3) as dsRNA-interacting antiviral proteins and show their requirement for type-I interferon induction. Depletion of TAO kinases in mammals or flies leads to an impaired response to virus infection characterized by a reduced induction of interferon stimulated genes in mammals and impaired expression of *srg1* and *diedel* in flies. Overall, our study shows a larger set of proteins able to mediate the interaction between viral genetic material and host factors than anticipated so far, attesting to the ancestral roots of innate immunity and to the lineage-specific pressures exerted by viruses.

[1] Technical University of Munich, School of Medicine, Institute of Virology, 81675 Munich, Germany. [2] Université de Strasbourg, CNRS UPR9022, Institut de Biologie Moléculaire et Cellulaire, Strasbourg, France. [3] Innate Immunity Laboratory, Max-Planck Institute of Biochemistry, Martinsried 82152, Germany. [4] CeMM - Center for Molecular Medicine of the Austrian Academy of Sciences, 1090 Vienna, Austria. [5] Computer Science and Mathematics Department, School of Arts and Science, Lebanese American University, Byblos, Lebanon. [6] Aarhus University, Department of Molecular Biology and Genetics - Structural Biology, Aarhus, Denmark. [7] Center for Physiology and Pharmacology, Medical University of Vienna, Vienna, Austria. [8] German Center for Infection Research (DZIF), Munich partner site, Munich, Germany. ✉email: andreas.pichlmair@tum.de

The innate immune system is critical to mount an appropriate defense response against invading pathogens. Virus infections in vertebrates lead to transcriptional and translational regulation of antiviral proteins (e.g., interferon-stimulated genes (ISGs) such as MX1, PKR, and IFIT proteins), to the secretion of cytokines with instructive functions (e.g., type-I interferons, IL-6, IL-8, and TNF) and alterations in the expression of cell surface proteins (e.g., MHC molecules)[1]. Among the best-studied cytokines that are involved in antiviral immunity are type-I interferons (IFN-α/β), a class of cytokines that are rapidly produced after virus engagement and that leads to upregulation of hundreds of proteins in an autocrine and paracrine manner[2]. It is commonly accepted that the main pathogen-associated molecular pattern that leads to activation of the innate immune system is viral genetic material, namely viral RNA and DNA[3]. These nucleic acids (NAs) are delivered into cells upon virus infection and are amplified during viral replication. Specific pattern recognition receptors (PRRs), such as membrane-bound Toll-like receptors (TLRs), cytoplasmic RIG-I-like receptors (RLRs), cGAS, or AIM2-like receptors sense viral nucleic acids and lead to the expression of type-I and type-III (IFN-λ) interferons, which serve as messengers to induce expression of antiviral proteins[4]. IFN-α/β and -λ bind to their specific cell surface receptors and induce the synthesis of ISGs. Importantly, besides activating PRRs, viral NAs can also associate with and activate the product of certain ISGs. Such proteins can scavenge viral genetic material (e.g., IFIT proteins), they can serve as co-receptors for PRRs (e.g., LGP2 and DDX41) or activate their enzymatic activity after viral RNA engagement to modulate cellular machineries (e.g., PKR, 2′5′-OAS proteins)[3,5]. Thus, viral NAs trigger multiple effects in cells, many of which are mediated by individual proteins.

The current understanding of how viral NAs induce antiviral immunity involves different scenarios. For instance, viral infections deliver nucleic acids into compartments that are normally NA-free. Such compartments can be endosomes, which are surveyed by TLRs, or the cytosol, which is normally free of DNA and is monitored by cGAS and AIM2-like receptors. Moreover, eukaryotic NAs are often heavily processed and bear methylated nucleotides (e.g., N7 methylation on CAP, 2′O methylation on 5′ ribose moieties) that are co- or posttranscriptionally added. Lack of these modifications, through transcription by viral polymerases devoid of editing activity, leads to the accumulation of nucleic acid species that are chemically or structurally distinct from cellular nucleic acids. However, most viruses that successfully infect eukaryotes have evolved strategies to mimic these modifications or to block PRRs sensing the lack of these marks. Moreover, some viruses transport or induce NAs that serve as signaling molecules, which can activate proteins with antiviral properties, e.g. 2′5′ linked oligoadenylates (2′5′OAs) and cGAMP[6]. Besides viruses, many bacteria contain similar oligonucleotides that can be sensed by the innate immune system[7].

Intracellular antiviral defense systems are best characterized in mammals. However, discoveries of viruses in fossils underline the coexistence of these pathogens and their hosts over hundreds of million years indicating that cell-intrinsic defense mechanisms should be similarly ancient[8]. Indeed, even though the interferon system only evolved in vertebrates, functional studies in the model organisms Caenorhabditis elegans and Drosophila melanogaster demonstrated that certain aspects of antiviral immunity are conserved between mammals and invertebrates[9]. The D. melanogaster DEAD-box RNA helicase Dcr-2, for instance, is related to mammalian RIG-I and serves as an antiviral protein in flies[10]. The recent discovery of an antiviral DICER isoform (aviD), which is active in specialized mammalian cells, further underlines the conservation of antiviral mechanisms between vertebrates and invertebrates[11]. Another DEAD-box

RNA helicase, DDX17, also acts as a cytoplasmic viral NA sensor, and both DDX17 and its fly orthologue, Rm62, have notable antiviral activity against Rift Valley Fever virus[12]. Similarly, the conserved dsRNA-binding enzyme ADAR binds viral generated double-stranded (ds)RNA and exerts adenosine to inosine editing activity in worms, flies, and mammals[13–15]. Furthermore, identification of a functional orthologue of the human cytosolic dsDNA receptor cGAS in sea anemone shows that the cGAS-STING signaling pathway was already present >500 million years ago, predating the evolution of interferons[16]. The latter examples suggest that not only the editing of RNAs but also downstream functions linked to virus defense programs, have been conserved. Noticeably, proteins that proved to be useful for antiviral immunity are—at least in part—evolutionarily conserved and have retained ancestral properties.

Affinity proteomics using viral NAs as baits are commonly employed to identify individual proteins with distinct functions in antiviral host defense. For example, a recently published, in-depth study of RNA-binding proteins (RBPome) in Sindbis virus (SINV) infected cells identified 247 RNA-binding proteins with differential binding during infection[17]. Similarly, cross-linking methodologies allowed for the detection of early viral RNA-binding proteins during Chikungunya virus and Influenza A virus (IAV) infection and identified ~400 viral NA binding proteins per virus[18].

Here, we perform affinity purifications of 17 different NAs (11 baits and 6 controls) in three different species (human, mouse, and fly), using liquid chromatography-tandem mass spectrometry (LC–MS/MS). Orthologues of the identified proteins were computationally compared, allowing the identification of cross-species-conserved NA interactors that retained antiviral properties throughout species evolution. Depletion screenings with 89 selected mammalian candidates and 92 fly genes followed by challenging with four viruses in human cells and five viruses in flies in vivo allow for cross-species comparison of pro- and antiviral activities in individual orthologous proteins. Among proteins that were identified to have evolutionary conserved functions were dsRNA-binding TAO kinases, which we show to be essential proteins for induction of the antiviral immune response in flies and humans.

## Results

**Proteomic identification of NA-interacting proteins in different species.** In order to identify proteins associated with NAs in different species, we established an affinity purification mass spectrometry (AP-MS) approach that allows testing for specificity to chemical or structural components of the different baits. Synthetic NAs were coupled to beads and incubated with cell lysates to precipitate NA associating proteins, which were then analyzed by LC–MS/MS (Fig. 1a)[19]. The bait NAs were chosen to resemble NAs commonly found during viral infections and that are known to activate or be targeted by the innate immune system: synthetic double-stranded (ds)RNAs (poly(I:C) and poly(A:U)), 5′ modified in vitro transcribed dsRNA (dsRNA-PPP and dsRNA-CAP0) and 5′ modified in vitro transcribed single-stranded (ss)RNAs (ssRNA-PPP, ssRNA-CAP, ssRNA-CAP0, and ssRNA-CAP1) (Supplementary Data 1)[20–38]. As controls, we used matched NAs: poly(C), poly(U), dsRNA-OH, and ssRNA-OH. Additionally, we included interferon stimulatory DNA (ISD) (as DNA bait)[39], as well as RNA:ISD (as a DNA-RNA hybrid), both with ssISD as a control. Lastly, the second messenger 2′5′ oligoadenylate (2′5′OA) was used with ATP serving as control.

Using these 11 bait and 6 control NAs, we performed 234 individual affinity enrichments with lysates from humans (THP-1 cells), mouse (Raw264.7 cells), and D. melanogaster (total flies

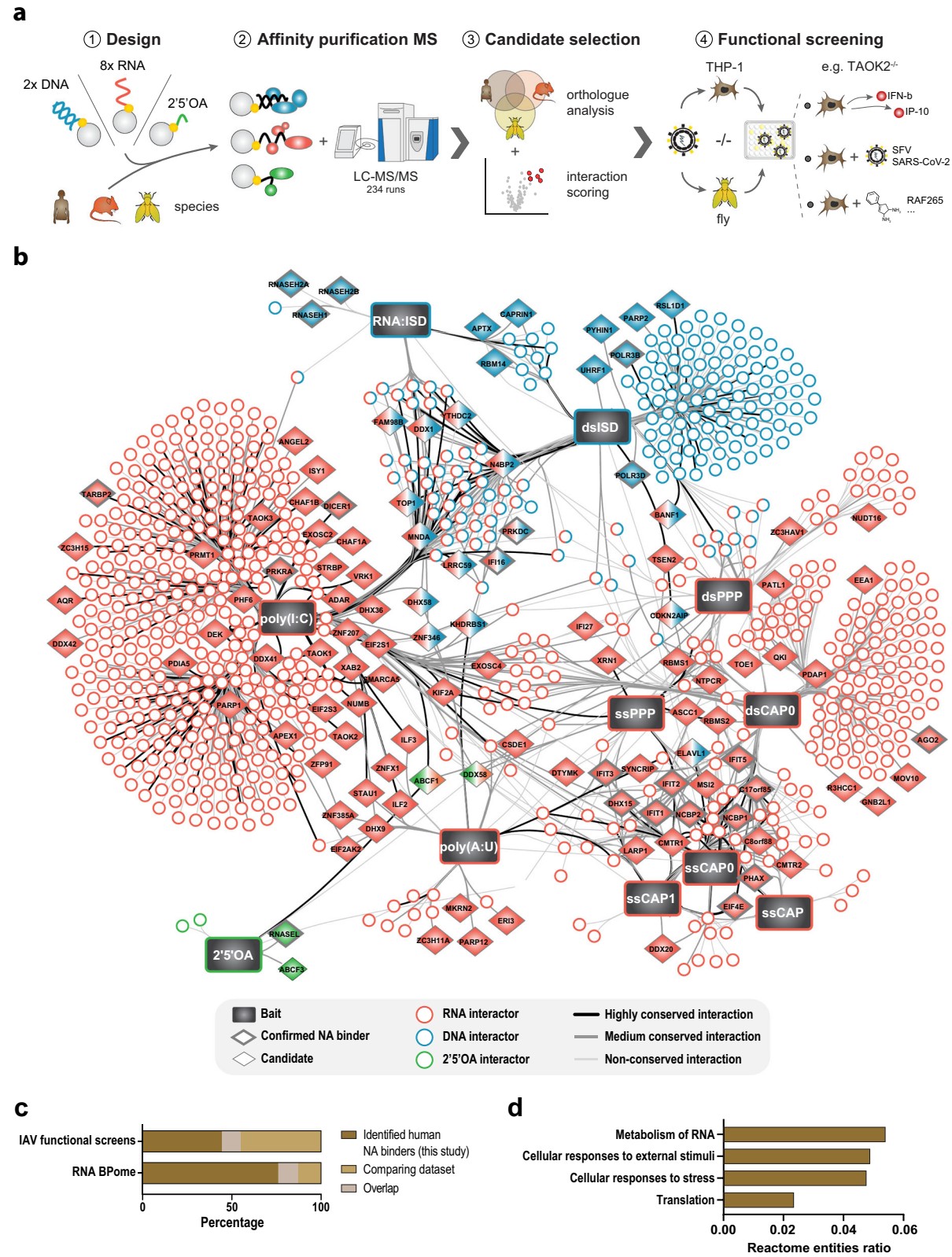

and Schneider S2 cells) origin. Overall, we identified 904 human, 1214 mouse, and 1479 fly proteins, respectively, which were significantly enriched in one or more affinity purifications (Supplementary Data 2–5).

The human dataset indicated an overall high specificity with expected binding patterns of known NA binders (Supplementary Fig. 1a). For instance, TREX1, which degrades IFN-stimulatory DNA (ISD)[3], bound specifically to all three ISDs; RNA:DNA hybrids recovered Ribonuclease H1, a known nuclease of RNA:DNA hybrids, as well as all three members of the ribonuclease H2 heterotrimeric complex (RNaseH2A, RNaseH2B, and RNase2C)[3]. In addition, an expected binding pattern was found for the known RNA-specific PRRs RIG-I and MDA5 and proteins of the IFIT complex, which are specifically associated

**Fig. 1 Proteomic identification of NA-interacting proteins in different species. a** The experimental workflow of the screen to identify NA-interacting proteins. NA baits were coupled to agarose beads and used to precipitate proteins from human, mouse, and fly cell lysates, followed by LC–MS/MS analysis. Analysis considering enrichment, regulation during immune responses, and cross-species conservation led to candidate proteins that were tested in functional screens in human cells and flies. **b** Network analysis of significantly enriched human proteins (Welch's *t*-test FDR <0.05) for each bait using THP-1 cells. Confirmed NA binders, selected candidates and conserved interactors are indicated, and proteins are colored according to the interacting NA type (red, RNA; blue, DNA; green, 2′5′OA). **c** Overlap between the enriched human NA binders and proteins identified with functional influence in loss of function screens testing replication of IAV (top)[41], and proteins changing their poly-A-RNA-binding pattern in SINV infected cells (bottom)[17]. **d** Results of the Reactome pathway enrichment analysis across all significantly enriched proteins independent of bait depicting the top enriched pathways (lowest FDR and highest entities ratios as defined by the number of identified proteins per pathway compared to the number of known proteins in the said pathway). Source data are provided as a Source Data file.

with chemically modified RNAs, including triphosphorylated and capped RNA baits[3]. The sensitivity and specificity of this approach was further supported by precipitation of RNase L only with its known ligand 2′5′OAs[6]. We further validated the AP-MS dataset by western blot confirming the NA interactions for selected candidates including SMARCA5 binding to poly(I:C) and PARP12 binding to poly(A:U) (Supplementary Fig. 1b). Interestingly, ABCF1 and ABCF3 bound to 2′5′OAs but not to ATP-loaded beads or dephosphorylated 2′5′OAs suggesting a phosphate-dependent interaction, such as known for RNase L (Supplementary Fig. 1b).

**NA interactome enriches for functionally relevant proteins.** Across all bait/control comparisons, we identified a total of 904 human proteins that were enriched for one or more baits (Supplementary Data 2). Network analysis of the obtained data showed similarities and specificities of binding patterns for each bait (Fig. 1b). In this network, prey proteins could be discriminated by the nature of the bait they associate with. For instance, RNA and DNA containing baits are stratified in two different topologies in this network. Furthermore, structural baits (e.g. dsRNA) could be discriminated from baits with chemical modifications such as triphosphates and RNA-cap modifications. In line with their structural similarity, ssCAP1 and ssCAP0 shared a large number of interactors ($n = 43$). A limited number of proteins ($n = 74$) are associated with both RNA and DNA baits. Among them was CDKN2AIP, which has been described as a regulator of DNA-damage signaling through p53[40] and was identified in genome-wide screens to modulate the growth of IAV, which generates PPP-RNA[41].

We intersected the binding patterns identified with RBPome in SINV infected cells[17] and identified 96 proteins that are also present in our human NA interactome dataset (Fisher exact test, *p* value: 2.42E-36) (Fig. 1c and Supplementary Data 6), in particular in the fraction of proteins precipitating with RNA. A cluster of proteins that was identified as AP-MS interactors and in the SINV RBPome includes MATR3, SFPQ, PSPC1, NONO, and RBM14. These proteins are members of the paraspeckle complex[42], which has been linked to antiviral immune responses[43,44]. SFPQ, NONO, and MATR3 have been shown to regulate posttranscriptional HIV-1 replication, with NONO directly interacting with both the HIV capsid and cGAS to impact the IFN response[45–47]. Our data indicates that these proteins interact with viral RNA and may thereby participate in antiviral immunity.

We also intersected the AP-MS data with a meta-analysis of genome-wide siRNA depletion datasets assessing IAV growth[41] (Fig. 1c and Supplementary Data 6). The overlap between NA interactors and genes identified as IAV modulators recovered 150 proteins, which represents a highly significant enrichment of functionally relevant proteins (Fisher exact test, *p* value: 2.23E-11). Of these, 134 are host factors, 12 are restriction factors, and four are noted as both host and restriction factors (KHSRP,

CIRBP, RRP1B, and PPAN)[41]. Of these proteins, six were previously annotated as NA binding factors with well-established antiviral activity (DHX15, PRKRA, EIF2AK2, IFIT2, POLR3B, and IFIT5). For example, PRKRA was enriched for the poly(I:C) bait in our affinity purification screen and was noted as a restriction factor for IAV. Upon binding to dsRNA PRKRA activates PKR/EIF2AK2, a well-described viral restriction factor[15], and has been shown to interact with IAV polymerase acidic protein, via a yeast two-hybrid screen[48]. The majority of proteins have not been associated to NA binding and it is likely that the affinity to viral nucleic acid contributes to the functionality of these proteins.

**A diverse set of functional activities is associated with NA interactors.** We performed enrichment analyses to extensively assess functions that were associated with the identified NA-interacting proteins. 60% of them were annotated as RNA and/or DNA binding (478 RNA binding, 147 DNA binding, and 69 bindings both) (Supplementary Fig. 1c). Reactome pathway analysis[49] using all proteins significantly enriched during the AP-MS independent of bait identified 34 overrepresented pathways (FDR <5.5E-15) (Supplementary Data 6). Of these, the four with the highest number of identified proteins per pathway were metabolism of RNA, cellular responses to external stimuli, cellular response to stress, and translation (Fig. 1d). Other relevant pathways included pathways related to viral infection, in particular viral mRNA translation and influenza infection, further indicating that the proteins identified in the AP-MS analysis are relevant both for NA binding and cellular response to infection. Since this dataset identified highly redundant GO terms we implemented a functional enrichment analysis that alleviates GO term redundancies (see Materials and Methods) (Supplementary Fig. 1d). This analysis recovered 27 enriched GO terms across the individual bait/control comparisons, 18 of which were directly related to NA processing. "Cellular response to exogenous dsRNA" was identified as hit for all RNA baits. Proteins contributing to the enrichment of this term included IFIT1 and RIG-I, as well as DHX9, all known NA binders with antiviral activity[3,50]. DNA-directed processes were more related to DNA containing baits, again confirming specificity. In particular, this enrichment analysis highlights association of proteins that are related to nucleic acid degradation. For instance, "nuclear-transcribed mRNA catabolic processes, nonsense-mediated decay", "mRNA stabilization", and "mRNA destabilization" were associated with many RNA baits indicating that viral RNAs are associating with mRNA processing related proteins. Proteins that are related to these terms include HNRNPR, MOV10, and DHX36 and highlight the prominent engagement of catabolic processes in antiviral immunity.

Enrichment of NA associating proteins was also reflected in enrichment for protein domains with annotated NA binding capability (Supplementary Fig. 1e and Supplementary Data 7). For example, the RRM superfamily (SF), an RNA-binding domain,

was enriched among proteins identified in affinity purifications of poly(I:C), ssPPP, ssCAP0, and dsCAP0. Similarly, we observed that the DNA binding domains homeodomain, bZIP, and HLH were all enriched in the dsISD affinity purification. This analysis led to some unexpected enrichments such as enrichment of the R3H domain, which has been annotated as specific to ssRNA/DNA binding, in a dsRNA containing bait, (dsCAP0), and HEAT EZ domain in ssRNA bait (ssCAP).

**Evolutionary conservation identifies antiviral proteins**. We next expanded the experimental approach to mouse (RAW263.7 macrophages) and *D. melanogaster* (total flies and Schneider S2 cells). In mouse and fly we detected 1214 and 1480 proteins, respectively, that were significantly enriched for one or more baits, with a similar distribution of enriched proteins for the individual baits (Supplementary Fig. 2 and Supplementary Data 3–5). The selective enrichment of proteins for RNA and DNA baits, which was also observed for the human system, suggested the similar quality of this dataset. The specificity of the dataset was supported by enrichment of individual proteins with known binding affinities. These included, for instance, enrichment of RNase L in 2′5′OAs, Rig in ds- and ssRNA, and OAS3 in poly(I:C) and poly(A:U) precipitates from mouse lysates. Similarly, we were able to identify a number of known NA interactors with reported antiviral activity in the fly (Supplementary Fig. 3 and Supplementary Data 4–5). Dcr-2, for example, was identified as enriched in poly(I:C) precipitates in both whole flies and S2 cells. Cpb20 and −80, components of the cap-binding complex, were binding all capped RNA baits used in both whole flies and S2 cells. Double-stranded RNA-binding proteins DIP-1 and Loqs were identified in precipitates containing dsRNA baits, but not when ssRNAs were used.

To identify proteins with species-conserved NA binding capability we compared human orthologues of the significantly enriched proteins in mouse and fly using the DRSC Integrative Ortholog Prediction Tool (DIOPT)[51]. Comparison of all enriched proteins, independent of bait specificity, identified 927 of the 2353 enriched proteins with a similar nucleic acid binding affinity between different species. Of the proteins enriched in the human affinity purification, 63% were also identified in mouse and 44% in flies, respectively (Fig. 2a and Supplementary Data 8–9). Comparing the affinity enrichment data for individual baits across species allowed us to identify conserved NA binders for each bait. For example, 127 proteins were conserved in the poly(I:C) precipitate, six in the ssPPP, and only one in 2′5′OAs (Fig. 2b). The single 2′5′OAs interactor conserved across all three species was ABCF1 that belongs to the ABCF subfamily of the ATP-binding cassette (ABC) transporter superfamily and has an AAA domain, which was identified as an enriched domain for 2′5′OAs interactors (Supplementary Fig. 1e). ABCF1 has previously been identified as an immune-regulatory protein in various cancer associated[52] and inflammatory conditions[53] and was more recently shown to function as an E2 ligase as well as a regulator of innate immune responses[54]. The association of ABCF1 to 2′5′OAs may allow the possibility to regulate the activity of this protein and therefore modulate inflammatory conditions. Moreover, our data indicate that 2′5′OAs may have additional intracellular targets and that 2′5′OAs may serve as signaling hubs to modulate the activity of proteins besides RNase L.

In order to select biologically relevant proteins, we calculated a score for each NA interactor. This score was based on the interaction strength between the bait and the protein in question, favoring less abundant proteins (see Materials and Methods). This was performed in parallel for the human and the mouse dataset, after which the 200 proteins with the highest score were

selected per bait and species. The highest-ranking proteins per bait were intersected between the two species and screened for being regulated by type-I and type-II interferons or association to GO terms related to NA sensing. Finally, this resulted in 90 candidates, which, based on their regulation and their affinity to NAs, showed a conserved pattern between mouse and human (Fig. 2c and Supplementary Fig. 4). Interestingly, many of the candidate proteins have only limited functional links to viral infection and immune regulation and were not previously known to interact with NAs (e.g., c8orf88, DTYMK, KIF2A, PDAP1, PDIA5, TAOK1, TAOK2, and VRK1).

**Antiviral properties of species-conserved NA associated proteins**. To explore whether the NA purification approach enriched for proteins with antiviral and/or immune-regulatory functions we conducted an arrayed lentivirus-based CRISPR/Cas9 knock-out (KO) screen in human THP-1 monocytes (Fig. 3a). To exclude toxicity effects associated with deletion of target genes, we performed a cell viability assay (Supplementary Fig. 5). Cells were left undifferentiated or were PMA-differentiated into macrophages followed by individual infection with luciferase-tagged vesicular stomatitis virus (VSV, non-segmented neg. strand ssRNA), Influenza A virus (IAV, segmented neg. strand ssRNA), Semliki Forest virus (SFV, pos. strand ssRNA), and Herpes-Simplex virus 1 (HSV-1, dsDNA), respectively. We tested the functionality of this screening method by depleting STAT1, a protein with known antiviral activity, as well as four nontargeting negative controls. As expected, compared to control depletion, luciferase signals for all viruses tested increased in *STAT1* depleted cells (Fig. 3b). Among the 89 candidates tested, depletion of 64 resulted in a significant change in luciferase activity for at least one virus in either the non-differentiated or differentiated cells (Fig. 3b). Depletion of 13 host factors led to reduced virus growth and 43 proteins served as restriction factors that led to increased luciferase levels upon depletion. In line with specificity for individual NAs, RNA-binding proteins had overall more prominent effects on RNA viruses.

This screen pointed towards a number of functional connections between NA affinity and specificity of antiviral activities. For instance, depletion of the PPP-RNA interactors *NTPCR*, *TSEN2*, and *RBMS2* allowed higher virus growth of the PPP-RNA generating IAV while these proteins did not affect other viruses tested (Fig. 3b). Moreover, in this screening system depletion of the DNA interactor *UHRF1*, a chromatin modifier linked to negative regulation of IFN-β expression[55], surprisingly promoted the growth of HSV-1, which could be in line with specific antiviral activity against DNA viruses and may call for further functional studies on this protein. Among proteins with a proviral activity, we identified the poly(I:C) and poly(A:U) associating proteins KHDRBS1, APEX1, and MKRN2 (associate with dsRNA and are proviral for SFV). Notably, MKRN2 has been shown to negatively regulate NF-κB through p65 upon LPS stimulation and depletion of *MKRN2* leads to increased IL-6 and TNF levels[56]. A similar function may be operative during viral infection explaining the decrease in viral load. Some proteins showed distinct activities depending on the virus tested. The poly(I:C) and PPP-RNA interacting proteins KIF2A and XRN1, for instance, appeared to be proviral for SFV, which generates high amounts of dsRNA, and antiviral for IAV, a PPP-RNA virus. The SFV proviral effect of XRN1 is in line with the phenotype observed during SINV infection, a virus belonging to the same family as SFV[17]. Our screen also provides insights into the differential requirement of proteins depending on the cellular differentiation state. Depletion of *DDX41*, a multifunctional protein and proposed DNA sensor and activator of the STING

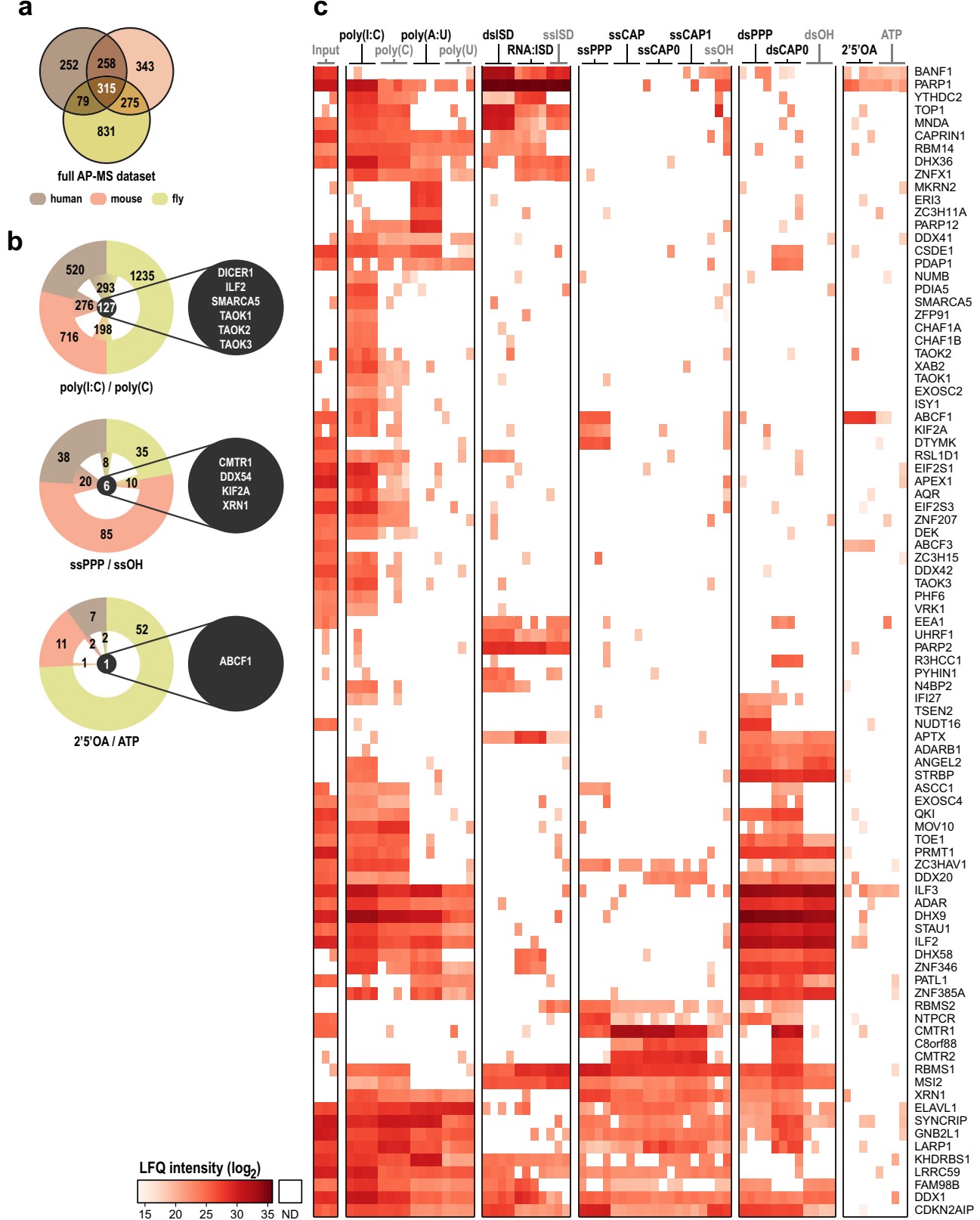

**Fig. 2 Conservation of NA binding across species and candidates selected for functional experiments. a** Venn diagram of all NA-enriched proteins identified in all AP-MS screens. **b** Significantly enriched proteins bound to poly(I:C) (top), ssPPP (middle), and 2'5'OA (bottom), including the overlap between species (brown: human, red: mouse, and green: fly). Proteins enriched in all three species are provided in the callout circle. **c** Heatmap showing log$_2$ LFQ intensities of 90 candidates individually selected for their specificity of enrichment for individual NAs, conserved enrichment between species, and regulation of protein abundance after virus infection, as well as protein abundance in the input lysate (Input). ND not detected. Candidate clustering is based on Euclidian distance and Ward as agglomeration method.

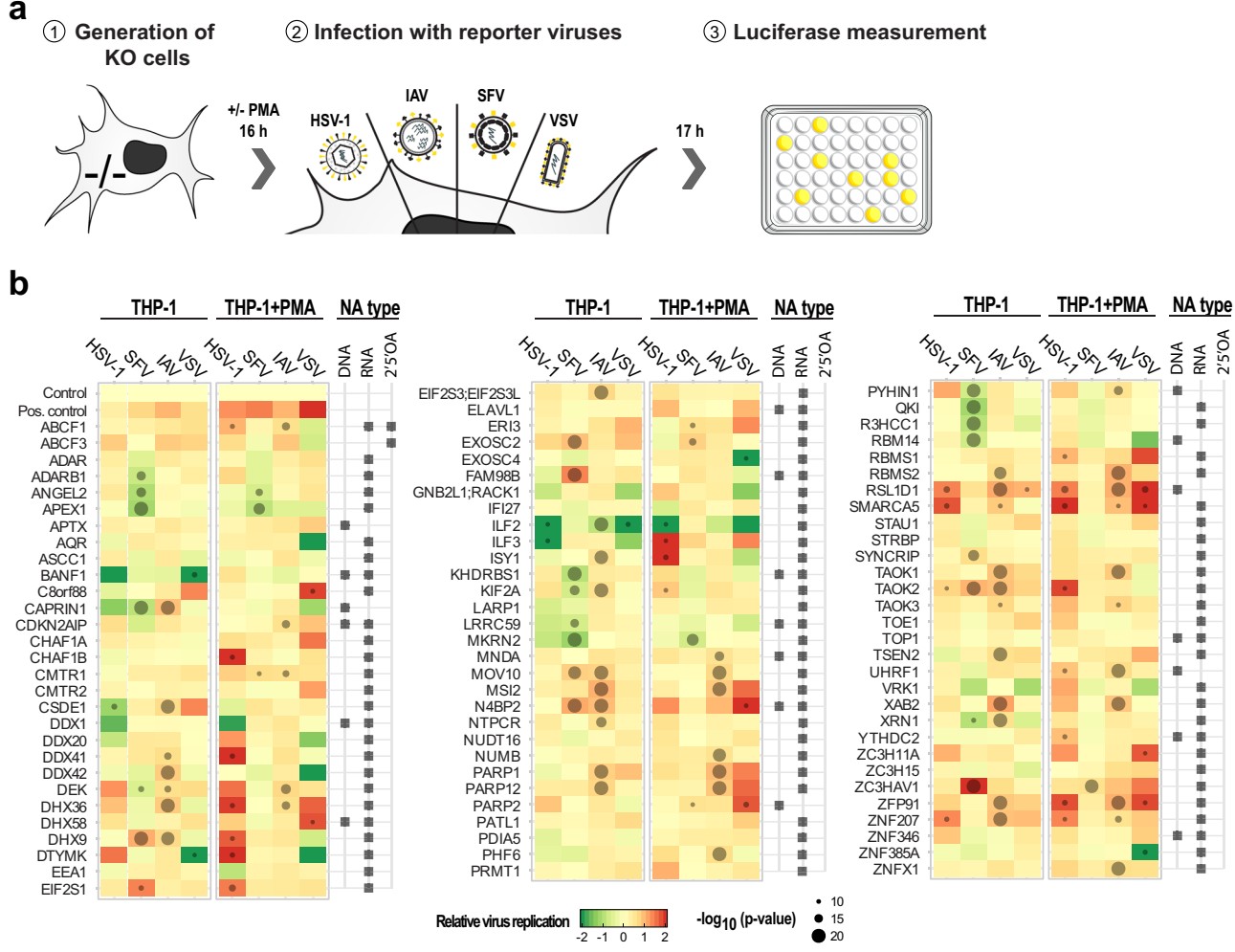

**Fig. 3 Antiviral activity of identified candidate proteins in human cells. a** Schematic of the screening strategy to test for virus-modulating activities of candidate proteins. THP-1 cells were infected with a pool of three lentiviruses expressing individual sgRNAs against the target protein or controls as well as CRISPR/Cas9 and selected for 16 days. Control: average of four pools of nontargeting sgRNAs; pos. control: *STAT1*. Cells were left undifferentiated or differentiated for 16 h with 150 nM PMA and infected with luciferase-tagged viruses (VSV-FLuc at MOI 0.1, IAV-GLuc at MOI 0.1, SFV-GLuc at MOI 0.1, and HSV-1-FLuc at MOI 0.2). After 24 h the accumulation of luciferase signal was analyzed. **b** Heatmap showing the $\log_2$ fold change of the luciferase signal in KO cells as compared to the control (median of the $\log_2(Luc_{KO}/Luc_C)$ posterior distribution). The two-sided *P* value is defined as the probability that $\log_2(Luc_{KO}/Luc_C)$ is different from 0 using a random-effects generalized linear Bayesian model; significant changes (*p* value ≤0.05; unadjusted for multiple hypothesis testing) are highlighted with dots. Data represents the median of biological triplicates. Candidates were further categorized into DNA, RNA, and/or 2′5′OA interacting proteins according to the results of the NA-AP-MS screen. Source data are provided as a Source Data file.

pathway[57], for instance, led to increase of HSV-1 growth in differentiated, but not in undifferentiated THP-1 cells (Fig. 3b). DDX41 is significantly associated with poly(I:C) and loss of the protein promoted growth of IAV (an RNA virus) in non-differentiated cells. Most interestingly, across this screen we identified proteins for which gene depletion led to generally increased virus replication, suggesting that they act as important regulators of the antiviral host-defense program. Prime candidates in this category were RSL1D1 (HSV-1, IAV, and VSV), ZFP91 (HSV-1, IAV, and VSV), and TAOK2 (HSV-1, SFV, and IAV). Overall, this NA interactor screen highlighted host factors with pro- and antiviral activity. These candidates were either specific to single viruses or more broadly active against viruses of different classes.

**Functional knockdown screen in *D. melanogaster* identifies proteins with a conserved antiviral function**. To enrich the functional dataset obtained in mammalian cells and to obtain

in vivo information from living animals, we conducted an shRNA knockdown screen in *D. melanogaster*. To this aim, we selected 92 proteins identified in the whole flies and S2 cells AP-MS screening. Of these, 69 were identified in both flies and S2 cells, 15 only in flies and 8 only in S2 cells (Supplementary Fig. 6), and 55 have a human orthologue (Supplementary Data 9). We used transgenic flies containing shRNA or inverted repeat transgenes under the control of a temperature-sensitive promoter, allowing temperature-inducible depletion of candidate genes (Fig. 4a). After activation of RNAi by temperature, the flies were infected with five well-characterized RNA viruses infecting arthropods (note that meta-genomic analysis demonstrated that *D. melanogaster* is essentially associated with RNA viruses)[58]. The replication of the Drosophila C virus (DCV, pos. strand ssRNA), Cricket Paralysis virus (CrPV, pos. strand ssRNA), Flock House virus (FHV, pos. strand ssRNA), SINV, and VSV for 2 or 3 days was monitored by RT-qPCR (Fig. 4b). Depletion of *Argonaute 2* (*Ago2*), an essential component of the RNAi pathway, and a nontargeting sequence (*mCherry*) served as positive and negative controls, respectively. Of the 92 fly

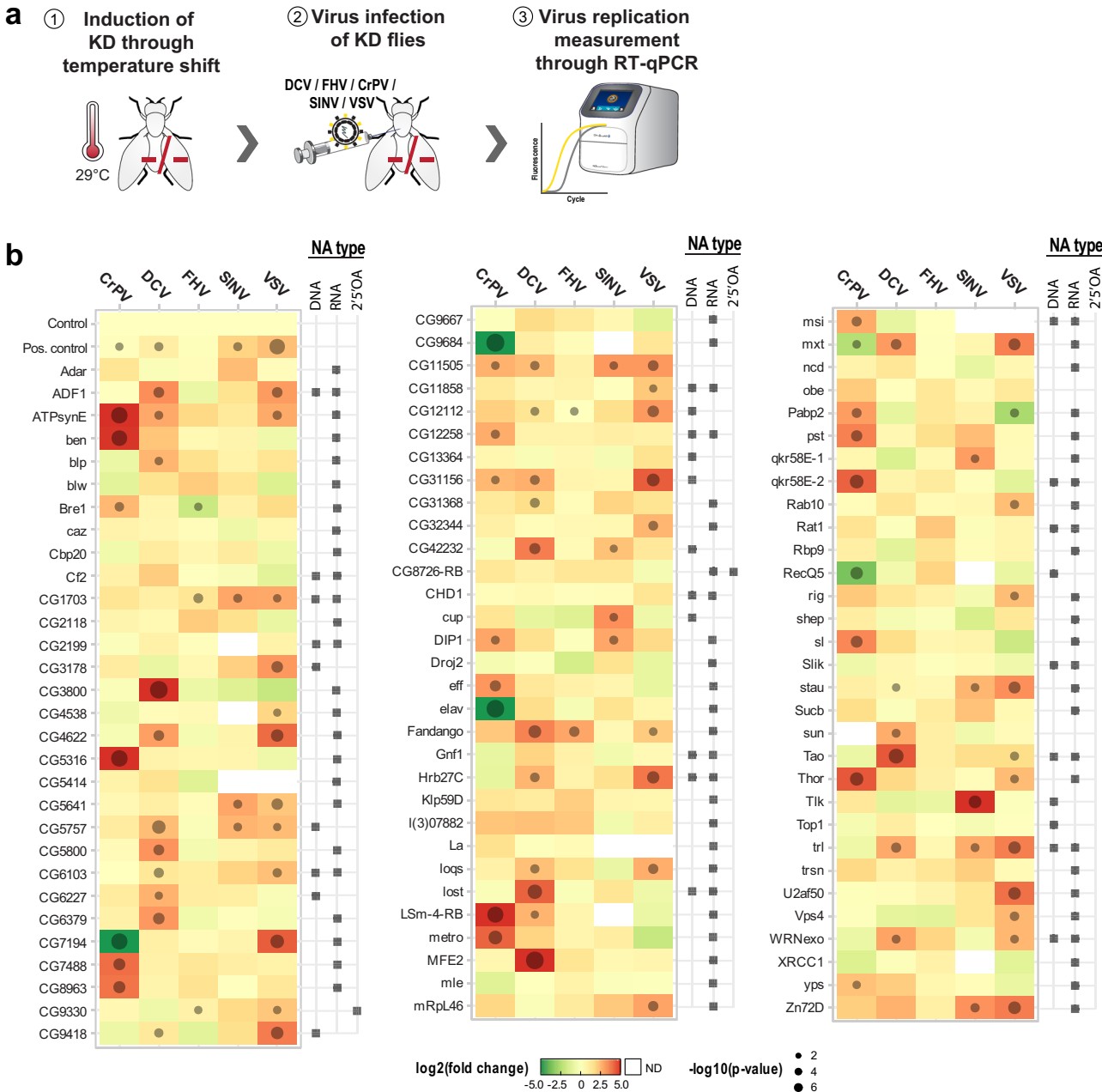

**Fig. 4 RNAi knockdown and virus replication in flies. a** Experimental procedure to test antiviral activity of candidate proteins in flies. The fly Gal4/Gal80[TS] system allows for the temperature-sensitive expression of sh or inverted repeat RNA. Flies carrying both the Gal4/Gal80 and UAS-siRNA were moved to 29 °C, activating Gal4 and inducing the siRNA KD. Upon confirming the viability of the KD flies, they were individually infected via injection of DCV (500 pfu/fly), FHV (500 pfu/fly), CrPV (5 pfu/fly), SINV (2,500 pfu/fly), and VSV (10,000 pfu/fly). Viral replication was measured by RT-qPCR at 2 (CrPV, DCV, FHV) or 3 (SINV, VSV) days post-infection. **b** Heatmap showing fold change of virus gene expression normalized to the housekeeping gene *RP49* upon target protein KD as compared to *mCherry* KD flies. Data represents the mean of biological triplicates. Significance was calculated using lsmeans R package (least-squares means, linear model) with Dunnet's adjustment for *p* values for multiple hypothesis testing. Candidates were further categorized into DNA, RNA, and/or 2′5′OA interacting proteins according to the results of the NA-AP-MS screen. Source data are provided as a Source Data file.

RNAi lines, 64 showed a significant change in viral load. Depletion of three host factors consistently reduced virus growth and 57 proteins served as restriction factors that upon depletion led to increased virus replication levels independent of the virus tested. Four candidate proteins were pro- or antiviral in a virus-dependent manner. The data allowed us to draw parallels between the NA association specificity and the viral replication phenotype of specific proteins. For example, qkr58E-1, which was identified as a poly(I:C) and poly(A:U) interactor had an antiviral phenotype in SINV, a virus known for producing large quantities of dsRNA.

Along the same line, Rig, which was precipitated with ssCAP0, ssCAP1, and dsCAP0, has antiviral activity against VSV, a virus generating capped RNA through its own capping machinery. As in the human CRISPR/Cas9 screen, we identified proteins with broad antiviral effects (stau, CG5757, CG11505, trl, ATPsynE, CG31156, fandango). Stau, which was identified as poly(I:C) interactor, is a known NA binder and has not been linked to viral infection in fly, though we observe an increase in virus replication for DCV, SINV, and VSV. Interestingly, a human orthologue of stau, STAU1, was upregulated during SINV infection[17] and has been shown to be

involved in the viral replication of Ebola virus, enterovirus 71, and IAV[59–61], further indicating that the intersection of fly and human data can provide insights into NA interactors involved in viral interactions.

**Cross-species interactome and functional KO screen identify TAO kinases as antiviral proteins**. The NA interaction and knockout/knockdown screens gave the opportunity to identify proteins that are functionally conserved across species and that are required for antiviral immunity. Comparison of the interactome in human, mouse and fly as well as the functional data obtained in human cells and flies, highlighted TAO kinases (TAOKs), a family of Ste20p-related serine/threonine kinases, as potential examples of antiviral proteins with conserved functions. The NA interaction screen shows specificity of TAO kinases to dsRNA since these proteins were enriched in poly(I:C) (in human, fly) and poly(A:U) (mouse) precipitates. The functional screen in human cells and in flies indicated antiviral activity of these proteins, supporting an evolutionary conserved function of TAOKs. Vertebrates, including mammals, amphibians, and fish, express three TAO kinases, while invertebrates, e.g., insects and nematodes, express a single TAO kinase. TAO kinases are characterized by an N-terminal serine/threonine-protein kinase catalytic domain and a largely unstructured region in the C-terminus. In addition, TAOK2 bears a transmembrane domain[62–64]. TAO kinases have previously been shown to regulate the p38 MAP kinase pathway upon UV-induced DNA damage through the phosphorylation and activation of MEK3/6[65]. Moreover, ectopic overexpression of TAOK2 was shown to activate apoptosis through activation of c-Jun N-terminal kinases[66]. Interestingly, an arginine to cysteine TAOK2 mutation in the unstructured C-terminus of the protein (R700C) was identified in clinical studies on patients suffering from generalized verrucosis, a human papillomavirus-induced disease[67], indicating a relevant link to virus infections. All three human TAO kinases showed antiviral activity against IAV infection, with TAOK2 also being antiviral against SFV and HSV-1 (Fig. 3b). The fly orthologue of all three human TAO kinases, Tao, was antiviral against DCV and VSV (Fig. 4b). Tao is an essential gene in *D. melanogaster* and its silencing led to a lethal phenotype within 3–14 days (Supplementary Fig. 7a), which did not allow long-term virus challenge experiments in vivo.

To validate AP-MS results, we applied co-immunoprecipitation followed by western blotting confirming the association of human TAO kinases with poly(I:C), but not poly(C), and verifying the requirement of dsRNA in this interaction (Fig. 5a). The control protein β-actin did not associate with poly(I:C) or poly (C), respectively. To elucidate whether the interaction between the TAO kinases and poly(I:C) is direct, we generated recombinant *D. melanogaster* Tao kinase (dTao) in insect cells and examined its interaction with poly(I:C) in a fluorescent quenching assay using a microscale thermophoresis. Indeed, dTao is associated directly with fluorescently labeled poly(I:C) with a $K_D$ in the nanomolar range ($42 \pm 15.66$ nM) (Fig. 5b), while denatured dTao did not show any interaction (Supplementary Fig. 8a). To test whether the kinase activity of dTao may be affected by dsRNA binding we evaluated its kinase activity in vitro in the presence or absence of poly(I:C). Notably, the addition of poly(I:C) led to a significant increase of dTao activity (Fig. 5c). Poly(I:C) itself did not affect the activity of the kinase assay. These data indicate that dTao kinase activity is modulated by poly(I:C), indicating a functional consequence of this interaction.

**Loss or inhibition of TAOK2 leads to a reduction of ISG expression and increases viral growth**. To gain a deeper

understanding of the functionality of TAO kinases in the context of virus infection, we applied full proteomic analysis using SFV-infected wild-type (wt) and *TAOK1*, -2, or -3 KO THP-1 cells (Supplementary Fig. 8b). These analyses allowed to evaluate the protein expression patterns of 5272 proteins in parallel and to assess the influence of TAOKs on the proteome. Compared to controls, depletion of all TAOKs led to significantly changed protein expression profiles after SFV infection with the most prominent effect being observed after depletion of *TAOK2* (Fig. 5d–f and Supplementary Data 11). In particular, in *TAOK2* KO cells, we could observe a decreased expression of proteins involved in the antiviral immune response, such as MX1, MX2, OAS3, and IFIT1, which was accompanied by an increased abundance of viral proteins (Fig. 5d). Similar regulation of MX1, as well as viral protein expression, was also observed for SFV-infected *TAOK1* and *TAOK3* KO cells (Fig. 5e, f). Indeed, GO term analysis based on differentially regulated proteins in SFV-infected *TAOK2* KO cells showed a enrichment for the terms "cellular response to type-I interferon" and "type-I interferon-mediated signaling pathway" (Fisher exact test, Benjamini–Hochberg FDR <0.05). Unbiased upstream promoter analysis[68] performed on all proteins that failed to be upregulated in SFV-infected *TAOK2* KO cells as compared to control cells, further indicated eight potentially linked transcription factor binding sites, including the ones for STAT1, IRF1, and STAT2 (Supplementary Fig. 8d). We independently validated the proteomics analysis testing for accumulation of SFV RNA and MX1 in THP-1 cells lacking TAOK1, -2, -3 or the control protein STAT1 (Fig. 5g, h). Indeed SFV RNA accumulated to significantly higher levels in cells lacking TAOKs with a particular increase in *TAOK2* deficient cells (Fig. 5g). At the same time, *MX1* mRNA transcripts were significantly reduced upon individual KO of all three TAOKs (Fig. 5h). Of note, even though the depletion of TAO kinases showed a prominent effect, the depletion of individual TAO kinases did not reach the same magnitude of effect as compared to *STAT1* depletion, which could be due to redundant effects of the individual TAO kinases. We evaluated whether the function of TAO kinases is conserved in *Drosophila* using an in vivo virus challenge model. To this aim, we infected control flies or flies with two different shRNAs targeting *Tao* with DCV for 2 days and monitored for expression of the virus response genes *srg1* and *diedel*. As expected, compared to control injections, expression of both genes was significantly induced in wt flies injected with DCV (Supplementary Fig. 7b, c). Notably, we could not observe significant induction of *srg1* or *diedel* in the two *Tao* knockdown flies. In conclusion, this analysis indicated that the function of TAOKs is conserved between invertebrates and vertebrates and that TAOK2 is particularly critical for antiviral protein expression and/or antiviral signaling cascade regulation in human cells.

**TAOK2 is involved in cytokine induction in response to SFV**. To further study the ability of TAOK2 in inducing innate immune responses, we confirmed its antiviral function against SFV expressing mCherry (SFV-mCherry) using fluorescence time-lapse microscopy (Fig. 6a). Quantification of the fluorescence signal showed that during the initial stages of infection, loss of TAOK2 and STAT1 lead to a comparable increase of virus production, which throughout the experiment stayed significantly higher than in control cells, confirming a prominent antiviral activity of TAOK2. Indeed, western blot analysis of SFV-infected *TAOK2* KO and *STAT1* KO cells indicated undetectable levels of MX1 while this protein was highly induced in control cells (Fig. 6b). To evaluate whether the inability to induce MX1 is due to IFN-α/β induction or type-I IFN signaling, we stimulated *TAOK2* KO, *STAT1* KO, and control cells with recombinant type-I interferon (IFN-α B/D). As expected, *STAT1* deficient cells did

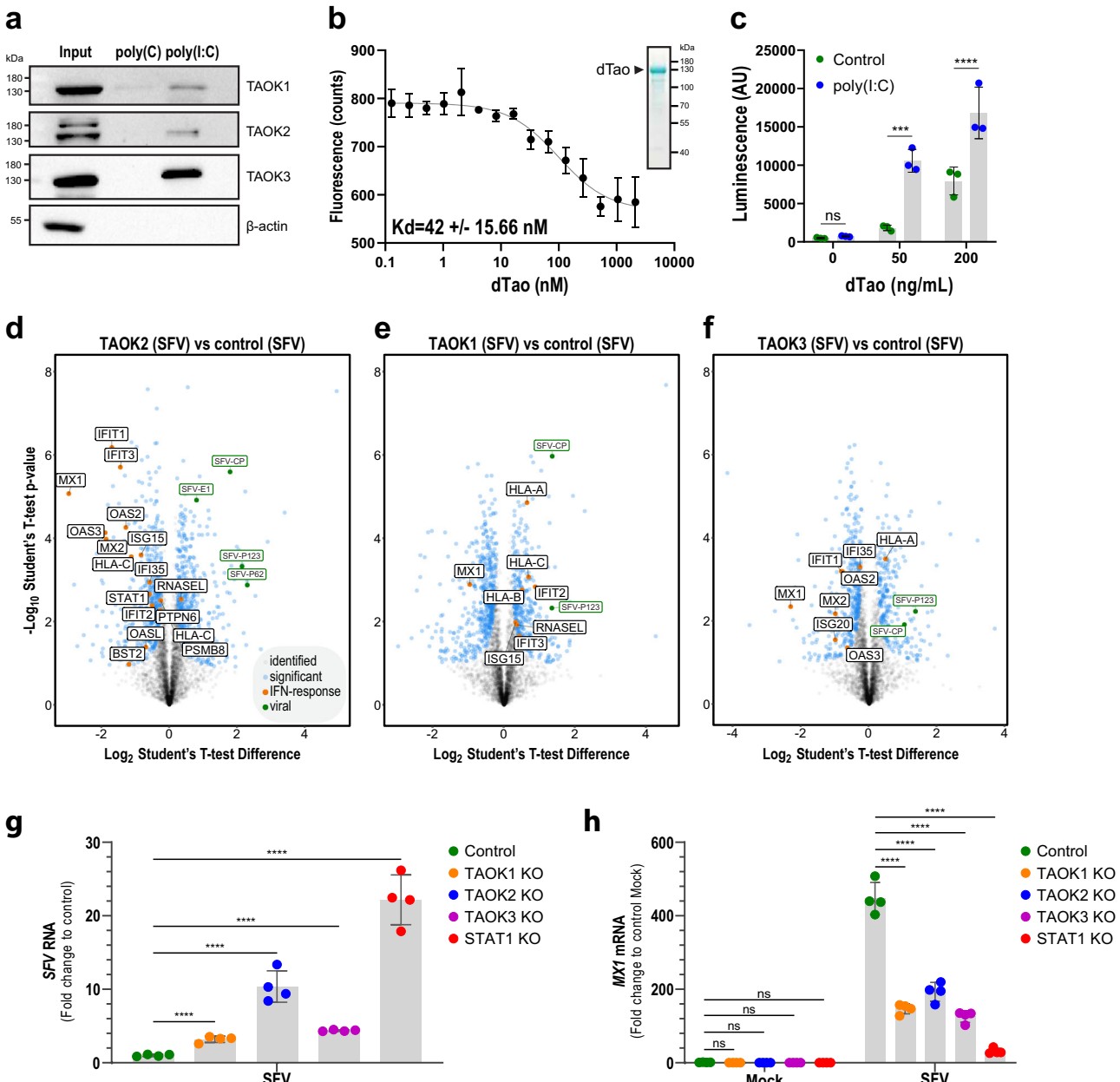

**Fig. 5 Drosophila Tao activity is regulated by poly(I:C) binding and human TAOK2 affects ISG expression. a** THP-1 cell lysate was incubated with poly(I:C) or poly(C) agarose beads and input proteins or co-precipitating proteins were analyzed by western blotting against the indicated proteins. **b** Fluorescence quenching assay showing fluorescence elicited from FITC-tagged poly(I:C) in presence of increasing concentrations of dTao. Shown are the mean fluorescence intensity ±SD of three measurements. The indicated $K_d$ was determined using Affinity Analysis v2.2.4 (NanoTemper Technologies). **c** Increasing amounts of dTao (50 and 200 ng/ml) were incubated or not with 0.3 mg/ml poly(I:C) and kinase activity was measured by a luminescence-based ATP consumption assay. Bars show the mean of three independent measurements ±SD. ****$p < 0.0001$, ***$p < 0.001$, ns $p > 0.05$ (Two-way ANOVA with Šídák's multiple comparison test). AU arbitrary units. Data presented in (**a**, **b**, **c**) is representative of at least three independent experiments. (**d**, **e**, **f**) THP-1 KO or nontargeted control cells were seeded and infected with SFV (MOI 10) for 24 h and analyzed for proteome expression. Volcano plots show protein expression patterns of SFV-infected *TAOK1* (**e**), *TAOK2* (**d**), and *TAOK3* (**f**) KO cells as compared to controls. Proteins with significantly different expression patterns are highlighted in blue, proteins belonging to the GO Term "cellular response to type-I interferon" are marked in orange and viral proteins in green. Data presented in (**d**, **e**, **f**) is averaged across four biological repeats; a two-sided Student's *t*-test (S0 = 0.1, permutation-based FDR <0.05) was used to assess the significance. **g**, **h** THP-1 KO and nontargeted control cells (green: control, orange: *TAOK1* KO, blue: *TAOK2* KO, purple: *TAOK3* KO, red: *STAT1* KO) were seeded and infected with SFV (MOI 1) or left uninfected (Mock) for 24 h and analyzed for SFV RNA and *MX1* mRNA transcript levels by RT-qPCR. Shown are the transcript levels normalized to the expression of the housekeeping gene *GAPDH* for four independent repeats expressed as fold change compared to the averaged biological repeats of the control. Error bars show the mean ± SD. ****$p < 0.0001$ (One-way ANOVA (**g**) or two-way ANOVA (**h**) both with Šídák's multiple comparison test). ns not significant. Data presented in (**g**, **h**) is representative of three independent experiments. Source data are provided as a Source Data file.

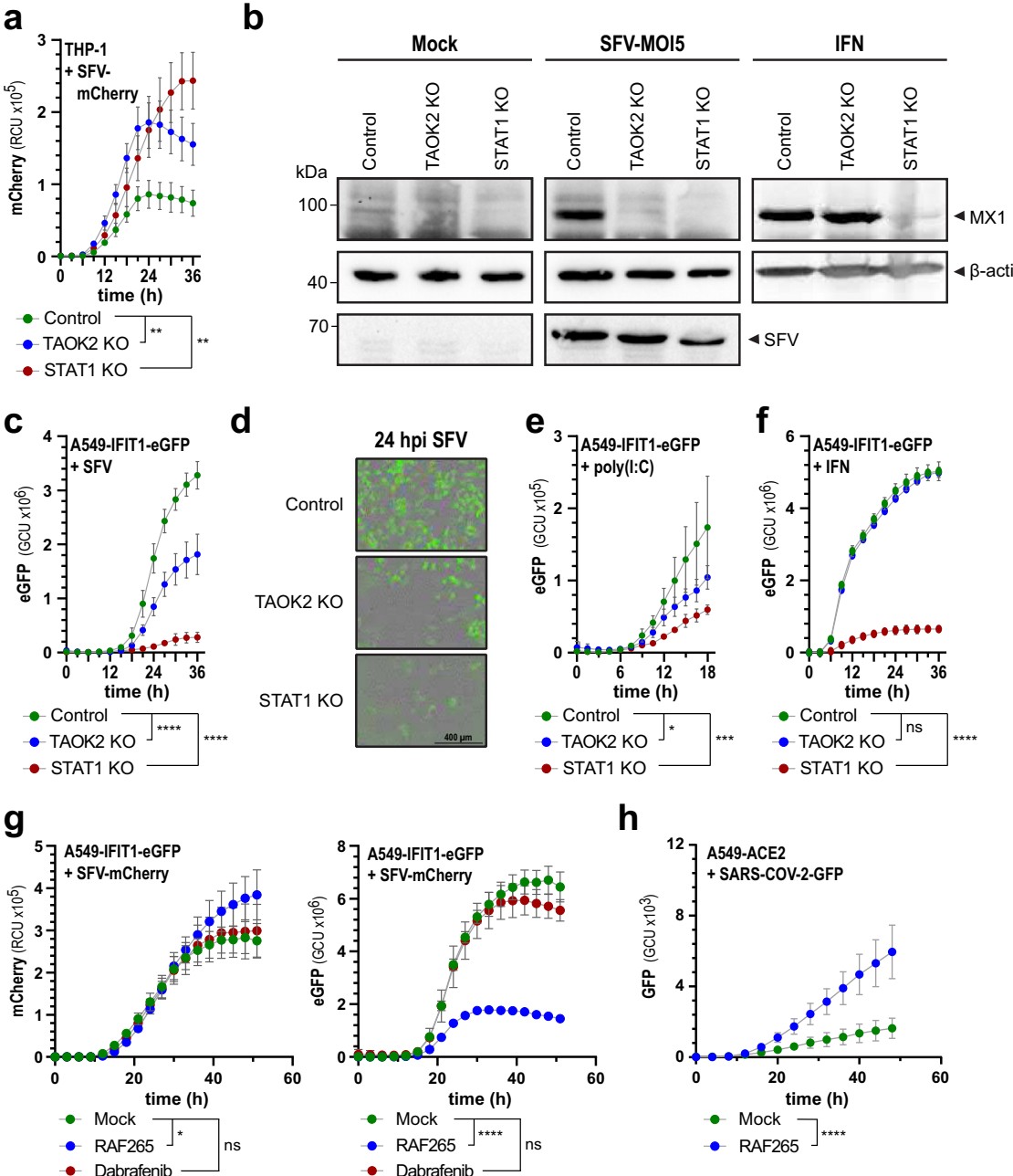

**Fig. 6 TAOK2 is required for IFN induction in infected and poly(I:C) stimulated cells. a** THP-1 cells were infected with SFV-mCherry (MOI 5) and red fluorescence intensity was measured every 3 h using an Incucyte S3 live-cell imaging system. The line diagrams show the mean integrated red intensity/cell confluence per image (RCU) ± SD (y-axis) over time (x-axis). **b** Nontargeted control (Control), *TAOK2* KO, or *STAT1* KO THP-1 cells were left unstimulated (Mock) or stimulated with SFV (MOI 5) or IFN-α B/D (1000 units/mL) for 24 h and used for western blotting against the indicated proteins. **c** Nontargeted control (Control), *TAOK2* KO, or *STAT1* KO A549-IFIT1-eGFP cells were infected with SFV (MOI 5) and green fluorescence intensity was measured at the indicated time points using an Incucyte S3 live-cell imaging system. Mean green intensity/cell confluence per image (GCU) ± SD (y-axis) is shown over time (x-axis). **d** Representative fluorescence microscopy images of (c) 24 h after infection with SFV. **e, f** as (c) but transfected with poly(I:C) (2 µg/mL) (**e**) or stimulated with IFN-α B/D (1000 units/ml) (**f**). Control cells are colored in green, *TAOK2* KO cells in blue, and *STAT1* KO cells in red. **g** A549-IFIT1-eGFP cells were infected with SFV-mCherry (MOI 5) and simultaneously treated with the TAOK2 inhibitor RAF265 (500 nM, blue), BRAF inhibitor Dabrafenib (500 nM, red), or left untreated (green). Green (GCU) and red fluorescence (RCU) intensities were measured at the indicated time points using an Incucyte S3 live-cell imaging system. Mean green or red intensity/cell confluence per image (G/RCU) ± SD (y-axis) is shown over time (x-axis). **h** A549-ACE2 cells were treated with the TAOK2 inhibitor RAF265 (10 µM) and infected with SARS-COV-2-GFP (MOI 3) and accumulation of GFP was measured over time in an Incucyte S3 system. Line diagrams show the mean green intensity/cell confluence per image (GCU) ± SD (y-axis) over time (x-axis). Data presented in (**a–h**) is representative of three independent experiments and for each line diagram, the mean ± SD are plotted based on at least four biological repeats. ****p < 0.0001, ***p < 0.001, **p < 0.01, *p < 0.05, ns p > 0.05 (Repeated measurements two-way ANOVA with Šídák's multiple comparison test to compare control versus treatment conditions). Source data are provided as a Source Data file.

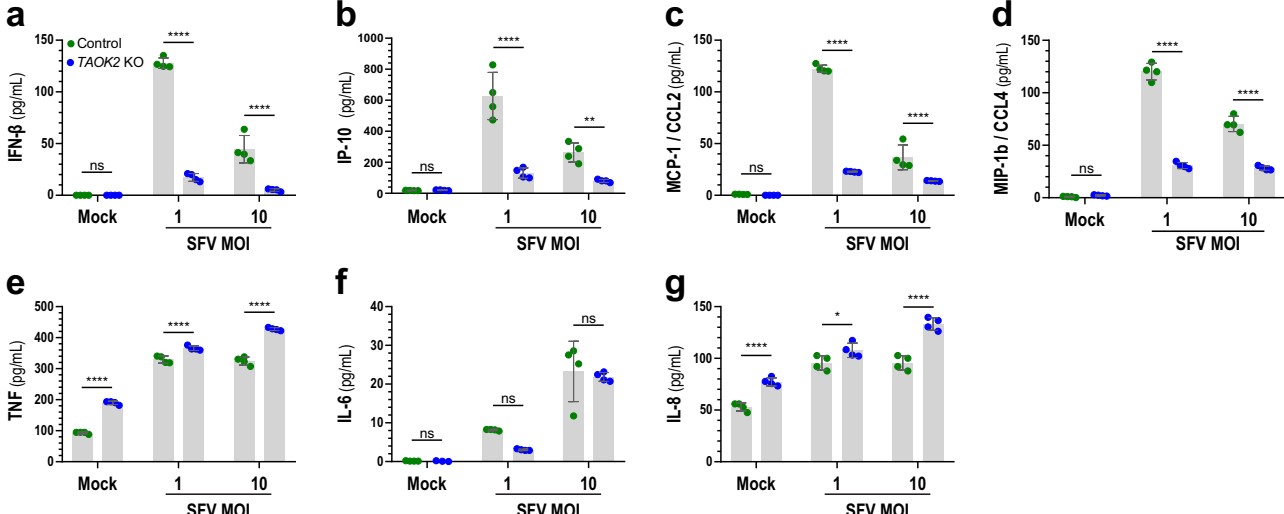

**Fig. 7 Loss of TAOK2 directly impacts IFN-α/β secretion. a–g** THP-1 control (green) or *TAOK2* KO (blue) cells were infected with SFV at the indicated MOI and 24 h later the accumulation of cytokines in the supernatant was measured by ELISA (IFN-β (**a**) and IP-10 (**b**)) and cytometric bead array (MCP-1/CCL2 (**c**), MIP-1b/CCL4 (**d**), TNF (**e**), IL-6 (**f**), and IL-8 (**g**)). Data presented is averaged across four biological repeats with mean ± SD. ****$p < 0.0001$, ***$p < 0.001$, **$p < 0.01$, *$p < 0.05$, ns $p > 0.05$ (Two-way ANOVA with Šídák's multiple comparison test). Source data are provided as a Source Data file.

not induce MX1, while expression of MX1 was similar in both *TAOK2* KO and control cells (Fig. 6b) indicating that signaling downstream of the interferon receptor is fully intact in *TAOK2* KO cells and that the failure to induce ISGs may be related to a defect in type-I interferon induction. To further corroborate these data and to gain additional quantitative and kinetic information on the induction of antiviral genes, we employed an A549-based reporter cell line that expresses GFP under control of the interferon-responsive *IFIT1* promoter (A549-IFIT1-GFP), which was depleted for *TAOK2* or *STAT1* (Supplementary Fig. 8c). Compared to control-targeted cells, *TAOK2* KO A549-IFIT1-GFP cells expressed significantly reduced amounts of GFP in response to SFV, confirming a defect in the type-I interferon induction or signaling (Fig. 6c, d). When using interferon-inducing poly(I:C) for stimulation experiments, we also found a significant requirement of TAOK2 to express IFIT1-driven GFP, indicating that the antiviral interferon system was affected by the loss of TAOK2 (Fig. 6e). Notably, wt and *TAOK2* KO A549-IFIT1-GFP cells responded similarly when stimulated with RIG-I activating triphosphorylated RNA (IVT4) or by transfection of the signaling molecule MAVS (Supplementary Fig. 8e), which indicates the specificity of TAOK2 to poly(I:C) and is in line with the affinity of TAOK2 to long dsRNA. As for THP-1 cells, IFN-α B/D treatment of A549-IFIT1-GFP cells led to a similar expression of GFP in both, control and *TAOK2* deficient cells (Fig. 6f), confirming that IFN signaling is not affected in *TAOK2* KO cells.

Kinases are commonly targeted for pharmacological interventions in a variety of diseases. We mined a kinase inhibitor-wide database for potential drugs that modulate the activity of TAOK2. A mass spectrometry-based screening approach identified RAF265 as a TAOK2 inhibitor[69]. RAF265 was originally identified as a BRAF inhibitor, and mouse experiments indicated potential use of RAF265 in cancer treatment[70], however, the BRAF status of patients did not correlate with treatment efficacy indicating that this drug has additional targets[71]. We applied RAF265 to A549-IFIT1-GFP cells and infected them with SFV-mCherry to study whether this drug would affect virus growth or influence the expression of GFP. RAF265 treatment led to a significant increase in SFV-mCherry growth, particularly at later times of infection (Fig. 6g). At the same time, IFIT1-GFP expression was reduced in RAF265 treated cells (Fig. 6g), similar to the phenotype previously observed

in the *TAOK2* KO cells. To exclude a potential effect of BRAF in this system, we treated A549-IFIT1-GFP cells with the established BRAF inhibitor Dabrafenib and infected these cells with SFV-mCherry. Dabrafenib did not affect SFV-mCherry growth or IFIT1-GFP expression, indicating that BRAF inhibition was not responsible for the phenotype observed upon RAF265 treatment. We evaluated whether the effect of RAF265 was dependent on TAOK2 by stimulating A549-IFIT1-GFP wt and *TAOK2* KO cells with SFV and monitored the effect of the inhibitor on IFIT1-GFP levels. While RAF265 significantly reduced IFIT1-GFP expression in wt cells, the inhibitor was much less active in *TAOK2* KO cells, further validating that RAF265 operates in a TAOK2 specific manner (Supplementary Fig. 8f). To underline the relevance of TAOKs upon infection with a human-relevant pathogen, we investigated the effect of RAF265 on SARS-CoV-2, a positive ssRNA coronavirus that generates large amounts of dsRNA during virus infection[72]. Treatment of A549-ACE2 overexpressing cells with RAF265 followed by infection with SARS-CoV-2-GFP showed an increase of viral replication compared to the negative control (Fig. 6h), indicating an involvement of TAOKs in antiviral immunity against a wide variety of viruses. Moreover, it indicates that the antiviral activity of TAOKs can be pharmacologically targeted.

**Loss of TAOK2 impacts IFN-α/β expression but has little effect on pro-inflammatory cytokines.** Having noted a distinct lack of upregulation of ISGs in SFV-infected TAOK2 deficient cells, but a functional ISG response to recombinant IFN, we hypothesized that TAOK2 is active within a PRR signaling pathway. Since PRR signaling results in cytokine expression, we assessed the expression of cytokines in supernatants of SFV-infected THP-1 *TAOK2* KO and control cells using a cytometric bead array and ELISA. Indeed, we observed a significant decrease in induction of IFN-β and IP-10 in cells that lacked TAOK2 as compared to control cells (Fig. 7a, b). A similar deficiency was seen for the cytokines MCP-1 and MIP-1b (Fig. 7c, d). MCP-1 and MIP-1b are both pro-inflammatory chemokines and chemo-attractants and induced via NF-κB and IRF3 activation or IFN-α/β stimulation, respectively[73,74]. Surprisingly, compared to controls, *TAOK2* KO cells did not produce less IL-6, IL-8, and TNF in response to SFV

(Fig. 7e–g), indicating that TAOK2 is not required to induce the expression of pro-inflammatory cytokines. The congruence of the lack of upregulation of ISGs and the decrease in IFN levels upon infection clearly indicates that TAOK2 is involved in the IFN production pathway and points towards the involvement of TAOK2 in IRF3-dependent cytokine expression (IFN-β and IP-10). Similarly, compared to control THP-1 cells, deletion of *TAOK1* and *-3* also led to a significant decrease of IFN-β and IP-10 protein secretion upon SFV infection (Supplementary Fig. 8g, h), indicating a nonredundant role of all TAO kinases for induction of IRF3-dependent cytokines.

**TAOK2 interacts with TRIM4, a known enhancer of RIG-I-dependent innate immune responses**. To identify a molecular link between sensing of dsRNA and the activation of innate immune responses, we used AP-MS to identify cellular binding partners of TAOK2[75]. To this aim we transfected HEK293T cells with control proteins (GFP and human PGAM5), C-terminal transmembrane domain deleted V5-tagged wild-type rat TAOK2 (which is highly similar to human TAOK2), rat TAOK2 D151A (mutation in the active site of the kinase domain), and rat TAOK2-R702C corresponding to human R700C (a clinical variation identified in human papillomavirus-driven generalized verrucosis). The transfected cells were either mock-treated or stimulated with poly(I:C), the V5-tagged proteins were precipitated and then analysed by mass spectrometry. Among the candidates that were identified to significantly associate with precipitated wt TAOK2 were proteins linked to already described functions of TAOK2, such as MAPK signaling (e.g., BRAP, PDCD10, WDR83, and YWHAB) and cell death (e.g., BAG1, BCLAF1, CHD8, STK3, and USP24). Intriguingly, besides expected proteins, TAOK2 specifically enriched for four functionally connected ubiquitin ligase proteins, TRIM4, TRIM21, FBXO30, and HECTD1 (Fig. 8a, Supplementary Fig. 9a, and Supplementary Data 12). While the co-enrichment of TRIM4, FBXO30, and HECTD1 could be detected in both mock and poly(I:C)-treated cells, their association to TAOK2 was more pronounced and only significant upon poly(I:C) stimulation. TRIM4, in particular, has been shown to be a critical mediator for IFN-α/β induction and is known to mediate the K63-linked ubiquitination of RIG-I[76] and could explain the effect of TAOK2 on type-I IFN induction in SFV-infected cells (Fig. 8b and Fig. 5d–h). The TAOK2 mutation in the active site of the kinase domain (D151A) greatly decreased the number of interactors, particularly association to proteins involved in the regulation of transcription (e.g., ASF1A, LRRFIP, and BEND3), but not to proteins related to MAPK signaling (e.g., CAMK2G, WDR83, and YWHAB) (Supplementary Fig. 9b). Notably, wt TAOK2 and TAOK2 D151A precipitated TRIM4 to a similar extent. Strikingly, the association of TRIM4 to TAOK2 was significantly reduced in the TAOK2-R702C mutant, the TAOK2 variant identified in immunodeficient patients suffering from uncontrolled generalized verrucosis (Fig. 8c)[67].

Overall, TAOK2 was identified in this study as a species-conserved dsRNA-binding protein involved in antiviral immunity. Lack of *TAOK2* significantly altered IFN-α/β induction, subsequent ISG expression and hence promoted viral growth. Unbiased proteomics analysis identified TRIM4, a known ubiquitin ligase with the ability to regulate RIG-like receptor signaling, as a prominent binding partner of TAOK2, which potentially links TAOK2 activation to the type-I IFN pathway.

## Discussion

The eukaryotic innate immune system coevolved under the selective pressure of viruses over millions of years, which resulted in conserved eukaryotic proteins dedicated to antiviral immunity[8,77]. We used affinity enrichment with viral NAs in distantly related species to identify conserved NA interactors, hypothesizing that these proteins may have preserved antiviral functions related to their NA interaction. These functions could include PRRs, like TLR3 and RIG-I, which detect viral NAs and induce cytokine expression, NA receptors with direct antiviral activity, e.g., PKR, which directly interferes with viral translation, or PRR cofactors, e.g., IFI16, which facilitates cGAS activity[3,78]. NA binding proteins might also have multiple or alternative functions, for example, the TLR7/TLR9 cofactor CD14 also enhances the cellular uptake of specific nucleic acids, promoting delivery to the respective TLR[79].

Affinity proteomics of NAs proved to be successful for proteins that bind viral NAs with high affinity, though it is less suited to identify functionally relevant proteins with limited affinity for viral NAs. For example, 5′ triphosphate RNA (PPP-RNA) associated with high affinity to IFIT proteins, which appear to act as scavenging factors[19]. The established PPP-RNA sensor RIG-I is identified with much lower enrichment scores. Moreover, an additional complication is that some proteins identified by affinity proteomics are not specific to a certain bait and also show binding affinity to a variety of different baits. Single AP-MS experiments, therefore, need to be carefully controlled and allow only partial insights into the specificities for individual viral NAs. Overall, we identified a set of conserved interactants, underscoring the ancient origins of antiviral innate immunity[80], together with a number of species-specific candidates, reflecting differences in the evolutionary trajectories of antiviral immune systems in organisms independently confronted to specific viruses[81]. Our results provide a useful resource for future functional studies and highlight the potential of the approach, which could be used with other species (e.g., other vertebrates, vector mosquitoes, and nematodes) to provide broader phylogenetic perspectives.

CRISPR/Cas9 mediated depletion of a selected subset of NA interactors demonstrated antiviral activity against different viruses tested, which could partially be explained by their distinct affinity to nucleic acids. Such an example may be PARP12, a member of the protein family Poly-ADP-Ribose Polymerases, which we identified as poly(A:U) interactors in humans and mouse. Its association with RNA may be explained by four CCCH-type zinc finger motifs, which are known to be involved in nucleic acid binding of other PARP proteins[82]. *PARP12* depletion led to increased replication of IAV in human cells and has previously been shown to be active against VSV, Murine gamma-herpesvirus 68 (MHV-68), Venezuelan equine encephalitis virus, and Zika virus[83–85]. Although some of the antiviral activity of PARP12 has been linked to its ability to specifically target virus proteins, such as degradation of Zika virus NS1 and NS3 proteins[83], the broad activity against many viruses may be explained by its affinity for nucleic acids.

Our data may not only contain proteins with direct links to viral genomic nucleic acids but also candidates that become active during virus infection or during the innate antiviral response. RSL1D1 (also known as CSIG) was in our study identified as a DNA binder and is known for its involvement in the DNA damage response (DDR) after UV irradiation[86]. Despite its restricted affinity for DNA, we observed a broad antiviral activity against DNA (HSV-1) and RNA viruses (IAV, VSV), but this could very well be explained by the RSL1D1 dependent induction of DDR since all the above viruses have the ability to induce DNA damage during the infection process[87]. RSL1D1 furthermore regulates translation of the signaling protein PTEN[88], which negatively regulates PI3K signaling and could thereby influence the antiviral response to RNA viruses[89,90].

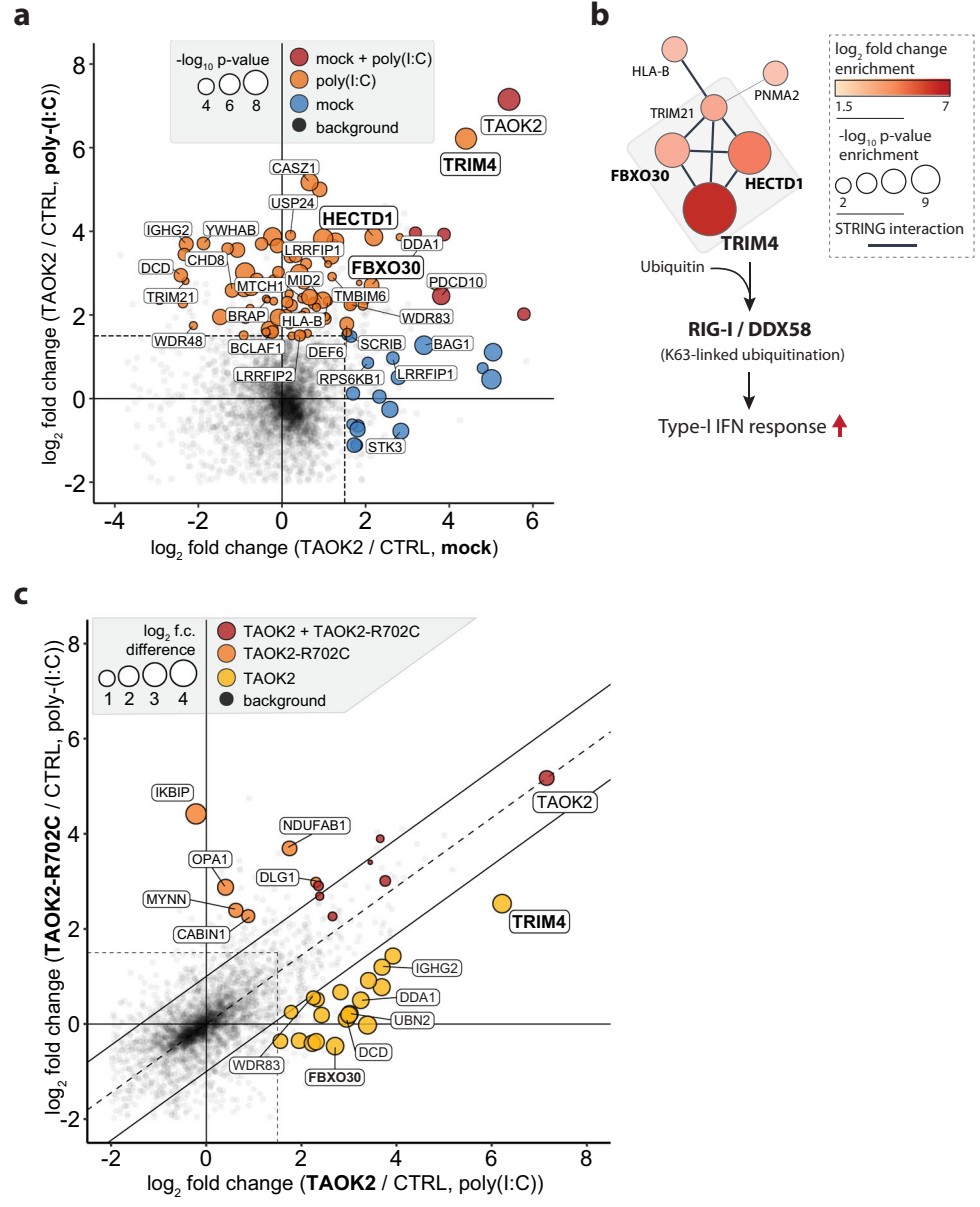

**Fig. 8 TAOK2 interacts with TRIM4, a known enhancer of RIG-I-dependent innate immune responses. a** Scatter plot showing the $\log_2$ fold change enrichment of proteins following affinity purification of rat TAOK2 as compared to the combined control (CTRL) baits (PGAM5 and EGFP) in mock ($x$-axis) and poly(I:C) stimulated ($y$-axis) HEK293T cells. Significantly enriched proteins were identified by a two-sided Student's $t$-tests (permutation-based FDR <0.05), further filtered to show a $\log_2$ fold change of ≥1.5, and colored in blue (only significant in mock), orange (only significant upon poly(I:C) stimulation), red (significant in mock and upon poly(I:C) stimulation), or black (nonsignificant or significant but a $\log_2$ fold change <1.5). Node size corresponds to the $-\log_{10} p$ value of a given bait versus control comparison and the node size for red-colored proteins is averaged between the two TAOK2 versus CTRL comparisons for mock and poly(I:C) stimulation. **b** STRING enrichment of rat TAOK2 interacting proteins in mock and poly(I:C) stimulated cells identified a functionally connected ubiquitin ligase complex, including TRIM4, an enhancer of type-I IFN responses by mediating RIG-I ubiquitination. **c** Scatter plot comparing the $\log_2$ fold change enrichment of proteins following affinity purification of wild-type rat TAOK2 ($x$-axis) versus rat TAOK2-R702C ($y$-axis) in poly(I:C) stimulated HEK293T cells. Significantly enriched proteins were identified by a two-sided Student's $t$-tests (permutation-based FDR <0.05), further filtered to show a $\log_2$ fold change of ≥1.5, and colored in yellow (only significant in wild-type TAOK2 with a $\log_2$ fold change difference ≥1 between wild-type and R702C-mutated TAOK2 affinity purifications), orange (only significant in TAOK2-R702C with a $\log_2$ fold change difference ≥1 between R702C-mutated and wild-type TAOK2 affinity purifications), red (significant in both wild-type and R702C-mutant TAOK2), or black (nonsignificant, significant but a $\log_2$ fold change <1.5, or significant and a $\log_2$ fold change ≥1.5 but an absolute $\log_2$ fold change difference <1 between wild-type and R702C-mutated TAOK2 affinity purifications). Node size corresponds to the absolute $\log_2$ fold change difference of a given protein between wild-type and R702C-mutated TAOK2 affinity purifications. $\log_2$ fold change difference values for each protein were normalized by the $\log_2$ fold change difference of TAOK2 to account for differences in enrichment efficiencies between the two TAOK2 variants.

Orthologue comparison of the human NA interactors to the interactors identified in fly and mouse provides information regarding evolutionary conservation and helps to pinpoint proteins for further investigation on a mechanistic level. Besides known conserved interactors, such as DICER1 (binding to poly(I:C)), a member of the RNA-induced silencing complex which posttranscriptionally silences genes both in humans and fly, we also identified less well-studied interactors as proteins with conserved affinity for viral NAs. For instance, both human and mouse SMARCA5 and their fly orthologue ISWI were identified in the poly(I:C) affinity purification. SMARCA5, a member of the SWI/SNF protein family, which is best known for its involvement in chromatin remodeling processes, showed an antiviral effect against VSV, IAV, and HSV-1 in human cells. While SMARCA5 has been studied in the context of cell invasion and migration in cancer[91], it has not yet been linked to antiviral immunity in humans. Interestingly, the mouse orthologue SMARCA5 was identified as a retroviral element silencer, and the fly orthologue ISWI has also been shown to interact with RNA and is upregulated during SINV infection[92–94]. We here identified SMARCA5 as an antiviral protein with conserved RNA-binding capability and antiviral activity, potentially pointing towards conservation of its function in distantly related species. SMARCA2 and SMARCA4, two other members of the SWI/SNF family, are differentially expressed during viral infection and regulate responses to poly(I:C)[95] and SMARCA2 impairs IAV growth[96]. In sum, this highlights how the SWI/SNF family could shape antiviral immunity while further studies may be warranted in the future.

Based on the NA affinity screen and candidate selection we identified 14 proteins that influenced virus growth in flies in vivo and in human cells in vitro (Fig. 9a). Some of these proteins shared affinities, functions, and known involvement in conserved signaling pathways. For instance, MSI2 and CDKN2AIP identified to interact with different ssRNAs and dsISD baits in fly and humans showed antiviral phenotypes for IAV and CrPV, and MSI2 additionally for DCV and VSV infection (Fig. 8b). Neither protein had previously been linked to a viral phenotype, but they are involved in the WNT-β-catenin signaling pathway, which has been linked to regulation of the IFN-β immune response in humans and to the Toll-regulated NF-κB response in fly[97,98]. Four of the 14 proteins with viral phenotypes in both fly and human (ADARB1, APEX1, ILF2, and KHDRBS1) showed contrasting effects in the functional screens. KHDRBS1 depletion reduced virus replication during SFV infection and is a proviral host factor for HIV-1, Foot-and-mouth disease virus, and Hepatitis C virus[99–101]. Surprisingly, KHDRBS1 depletion in flies led to an increase in viral replication for both CrPV and SINV. It is currently unclear how these opposing phenotypes occur in the two different species but may reflect differential adaptation of these viruses to the respective host[102].

Among the most conserved candidates in terms of binding capability and antiviral function were the three TAO kinases, TAOK1, TAOK2, and TAOK3, which interacted with poly(I:C) in human and mouse cells, while the single fly orthologue dTao interacted with poly(I:C) and dsISD. The interaction between dTao and poly(I:C) was direct and of high affinity and intriguingly, poly(I:C) regulated dTao kinase activity in vitro, indicating a direct consequence of this interaction. In functional screens, all three human TAO kinases showed antiviral activity against IAV infection, with TAOK2 also being antiviral against SFV and HSV-1. In flies, dTao was antiviral against DCV and VSV and its depletion led to reduced expression of virus-induced genes in vivo. Proteomics analysis showed a surprising involvement of TAO kinases in the induction of proteins involved in antiviral immunity. Differential expression of proteins in infected cells, as well as cytokine profiling, suggest that *TAOK2* deficiency leads to a selective impairment of IRF3-dependent cytokines such as IFN-β and IP-10.

Interestingly, recent single-cell genomics data performed in SARS-CoV-2 infected patients showed reduced expression of *TAOK1* in SARS-CoV-2 infected cells[103], indicating that this kinase may be actively regulated by viruses and further underlining TAOKs as important positive regulators in antiviral immunity. Notably, *TAOK2* mutations have also been identified in patients showing treatment-resistant generalized verrucosis lesions[67], a disease caused by the uncontrolled growth of human papillomavirus. This clinical report identified a missense mutation of *TAOK2* (C2098T), which causes an amino acid change (arginine to cysteine at position 700), in the unstructured region located between the kinase (28–281aa) and the transmembrane domain (955–1063aa). Even though papillomaviruses encode a DNA genome, it is well accepted that DNA viruses can generate dsRNA through convergent transcription which leads to activation of dsRNA-binding proteins such as PKR and ADAR1[104]. Our data suggest that the functional relationship between the *TAOK2* mutation and the observed inability to control the human papillomavirus could be related to the innate immune regulating properties of TAOK2 upon sensing accumulation of dsRNA.

As noted above, TAO kinases are highly conserved across species, including in the nematode *C. elegans*. In addition, potential orthologues have also been predicted in *N. vectensis* and *A. queenslandica*[105]. Looking at a broader evolutionary context, this suggests that TAO kinases may have evolved in parallel to TLRs and cGAS[16,106]. In the context of DNA damage, G-protein-coupled receptor signaling, and osmotic stress, TAO kinases are known regulators of the MAP kinases MKK3/6 and MKK4/7, which regulate p38 and JNK, respectively[65,107]. Both p38 and JNK have been linked to pro-inflammatory cytokine expression and interferon production[108]. Intriguingly, there is some evidence that the MKK4/7 – JNK signaling pathway directly and specifically regulates IFN production. For instance, upon loss of MKK4/7 poly(I:C) induced expression of IP-10 and IFN-β was reduced, while phosphorylation of the NF-κB regulator IκB was unaffected[109]. In line with this, treatment with the JNK inhibitor SP600125 inhibited poly(I:C) induced IRF3 phosphorylation and dimerization[110]. Activated TAOK2 may directly activate MKK4/7, and then specifically activate JNK and the cytokine response. It is also possible that TAOK2 forms a temporary complex with a PRR and viral NAs. Interaction with poly(I:C) increases TAOK2 activity, potentially leading to the phosphorylation of IFN-α/β inducing PRRs. Such a cofactor function would be comparable to the function of IFI16 in the cGAS-STING pathway[78]. We used AP-MS to identify intracellular binding partners of TAOK2. While we could not find evidence for the direct association of TAOK2 to pattern recognition receptors, we found that TAOK2 specifically interacts with TRIM4, an E3 ligase that has previously been shown to mediate K68-linked ubiquitination of RIG-I[76]. Notably, the association of TRIM4 to TAOK2 appeared to be more stable in poly(I:C) treated conditions and the single point mutation (TAOK2 R700C) identified in generalized verrucosis patients[67] reduced the association between TAOK2 and TRIM4. It is therefore possible that TAOK2 modifies the activity of a PRR-regulating protein. The consequence of this interaction remains to be further evaluated, but it may indicate yet unexplored functional relationships that could be relevant for antiviral immunity. Poly(I:C), which is binding and activating TAOK2, is sensed by the PRRs MDA5 and TLR3. TRIM4, which has originally been found to activate RIG-I by K68-dependent ubiquitination, may also regulate the activity of MDA5 or TLR3. An involvement of TAOK2 and TRIM4 in MDA5 dependent responses is further supported by the lack of IFN production in TAOK2 KO cells after infection with SFV, which is an MDA5

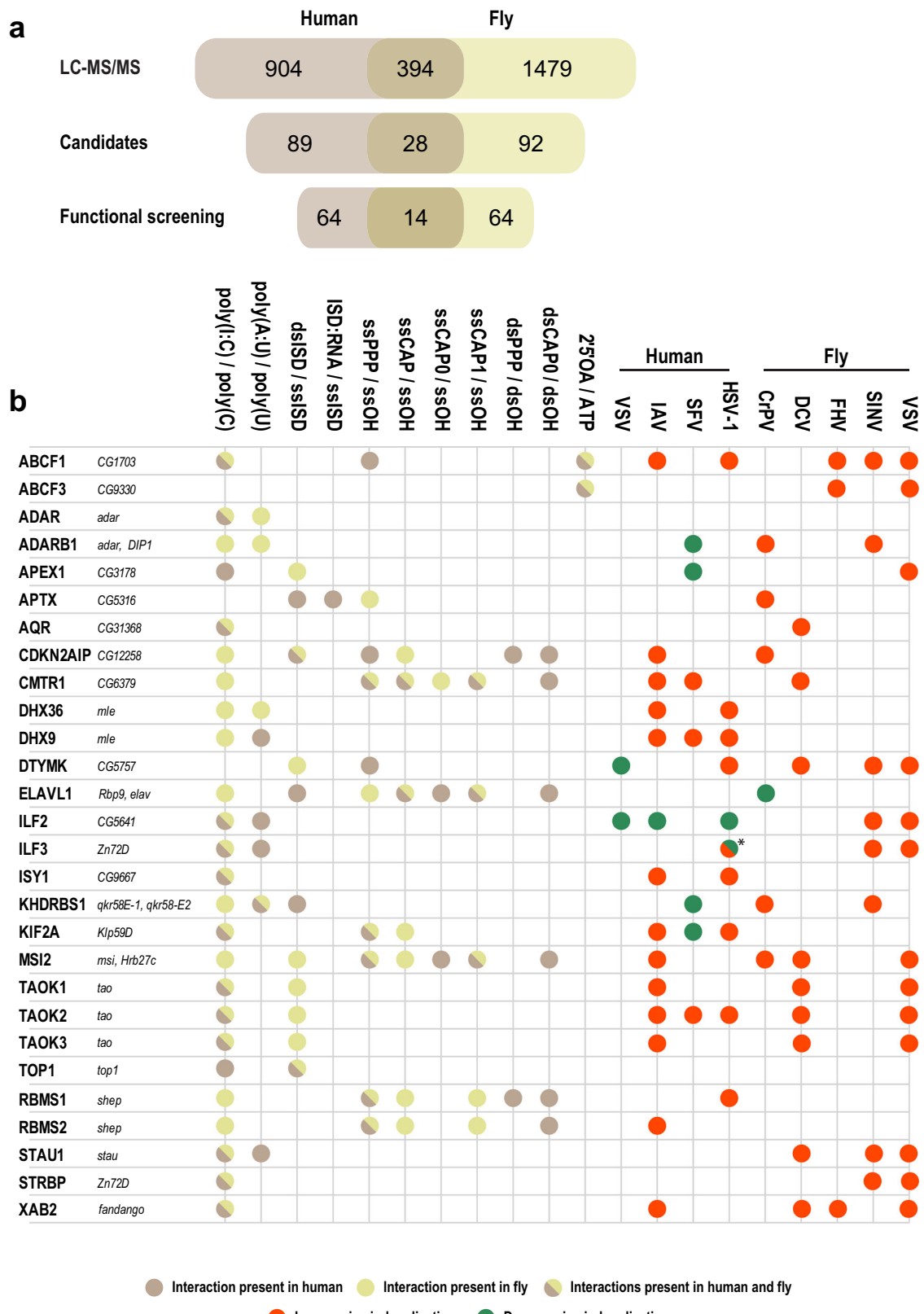

**Fig. 9 Summary of screening results. a** Overlap of proteins between human (brown) and fly (green; combined results from drosophila and S2 cells), which were identified as NA binder in the AP-MS screen (LC–MS/MS), selected as candidates for further functional validation (Candidates) and that showed an anti- or proviral phenotype upon depletion (Functional screening). Orthologue mapping between human and fly proteins was performed using DRSC Integrative Ortholog Prediction Tool. **b** Overview of the results gathered in the screening approaches in humans and fly for the 28 top candidates (red: antiviral, green: proviral). Shown are human proteins with annotated orthologues in fly. *for ILF3 we observed an increase of viral replication in PMA treated THP-1 cells, but a decrease of viral replication in the untreated THP-1 cells.

activating virus. Very little is known about the regulation of E3 ligases involved in innate immunity. However, one could speculate that TAOK2 regulates the activity of TRIM4 that in turn activate RIG-I, MDA5, or Toll-like receptor signaling through its E3 ligase activity.

Collectively our data show that the here described unbiased NA-AP-MS approach is suitable to identify yet unstudied proteins that are relevant for antiviral immunity.

## Methods

**Cells, flies, viruses, and reagents**. A549-IFIT1-eGFP cells were a kind gift from Ralf Bartenschlager (Heidelberg University, Germany), THP-1 cells from Veit Hornung (Gene Center Munich, Germany), A549 cells from Georg Kochs (University of Freiburg, Germany), RAW263.7 cells from Thomas Decker (MFPL Vienna, Austria), Schneider S2 cells from Irene Ferreira, and HEK293T cells were purchased from ATCC (CRL-3216). THP-1 cells were maintained in RPMI1640 (Sigma-Aldrich), A549, and RAW cells in DMEM (Sigma-Aldrich), both supplemented with 10% fetal calf serum (Sigma-Aldrich) and antibiotics (100 U/ml penicillin, 100 μg/ml streptomycin). If desired, THP-1 cells were differentiated with PMA (150 nM, Sigma-Aldrich P1585) upon seeding overnight before stimulation. S2 cells were maintained in Schneider's medium (Biowest) supplemented with 10% fetal calf serum, Glutamax (Invitrogen), and antibiotics (100 U/ml penicillin, 100 μg/ml streptomycin). Fly cultures were grown on standard cornmeal agar medium at 25 °C. All flies used were *Wolbachia*-free.

HSV-1 (F-strain)-F-Luc was a gift from Soren Riis Paludan (Uni Aarhus, Denmark), VSV-Luc from Gert Zimmer (University Bern, Switzerland), Influenza A SC35M NS1_2A_Gaussia_2A_NEP from Peter Reuther (University of Basel), SARS-CoV-2-GFP from Volker Thiel (Universität Bern), SFV6-2SG-Gaussia-Luc, and SFV-mCherry from Andres Merits (University Tartu, Estland). DCV was kindly provided by X. Jousset and M.Bergoin (INRA-CNRS URA2209, St Christol-Lez-Alès, France).

Firefly luciferase substrate Coelenterazine (C2230) was from Sigma-Aldrich. Poly(I:C) was purchased from Sigma-Aldrich (P9582), fluorescent poly(I:C) was purchased from InvivoGen (tlrl-picf). Transfection of nucleic acids and poly(I:C) was performed using Metafectene Pro (Biontex T040). RAF265 and Dabrafenib were both purchased from Cayman Chemical (CAYM16991-5 and CAYM16989-10, respectively). Recombinant human IFN-α B/D was a kind gift of Prof. Dr. Peter Stäheli.

The following antibodies were used: mouse anti Mx1 (1:1000) was a kind gift form Georg Koch (University of Freiburg, Germany), mouse anti actin antibody (1:2500) was purchased from Santa Cruz (sc-47778), anti TAOK1 from rabbit (1:1000) purchased from Bethyl Laboratories (A300-524A-M), anti TAOK2 from rabbit (1:500) purchased from Sigma-Aldrich (HPA010650), anti TAOK3 from rabbit (1:1000) purchased from Sigma-Aldrich (HPA017160), anti ABCF1 from rabbit (1:1000) purchased at Aviva Sytems Biology (ARP43631_P050), anti ABCF3 from rabbit (1:1000) purchased at Sigma (HPA036332), anti RNase L from mouse (1:2000) was a kind gift from Bob Silverman, anti RSL1D1 from rabbit (1:1000) was purchased from Sigma (HPA043483), anti SFV core from rabbit (1:1000) was a kind gift from Andres Merits, anti SMARCA5 from rabbit (1:1000) was purchased from Sigma (HPA008751), anti STAT1 from rabbit (1:1000) was purchased from Cell Signaling Technology (9172), horseradish peroxidase (HRP)-coupled secondary antibody against mouse IgG (1:2000) was purchased from Sigma-Aldrich (A0168) and against rabbit IgG (1:5000) was purchased from Cell Signaling Technology (7074). The human IFN-β DuoSet ELISA was purchased from Biotechne (DY814-05) and the human IP-10 OptEIA ELISA Set from BD Biosciences (550926). The Bio-Plex Pro Human Cytokine 17-plex was ordered from Bio-Rad (M5000031YV).

**NA affinity purification**. Synthetic oligoribonucleotides with a 3′-terminal C6 amino linker matching the first 22 nucleotides of the 5′ untranslated region of Severe Acute Respiratory Syndrome Coronavirus HKU-39849 [PPP-r(AUAUUAGGGUUUUUACCUACCC)-NH2] and a corresponding 2′O-ribose methylated RNA oligomer [PPP-r(AmUAUUAGGGUUUUUACCUACCC)-NH2] were ordered from ChemGenes Corporation (Wilmington, MA, USA) and capped as described previously using the m7G Capping System (CellScript). Capped RNA oligomers were then HPLC-purified, biotinylated with the biotin-*N*-hydroxysuccinimide ester (Epicenter) according to the manufacturer's instructions, and again HPLC-purified. As a control, we used a corresponding 3′-terminal biotinylated and HPLC-purified oligoribonucleotide harboring a 5′ hydroxyl group [OH-r(AUAUUAGGGUUUUUACCUACCCU)-biotin]. Biotinylated 2′5′OAs were synthesized according to the protocol described in Turpaev et al.[111], using a RESOURCE column, biotinylated ATP was purchased from Perkin Elmer (NEL544001EA). Biotinylated in vitro synthesized PPP-RNA (7SKas RNA) was described earlier[19]. For quantitative purification of proteins binding to biotinylated synthetic or in vitro transcribed nucleic acids, streptavidin affinity resin was first incubated either with 100 pmol aliquots of biotin-labeled 7SKas RNA, 5 nmol of RNA oligomers, 100 pmol 2′5′OAs, 100 pmol ATP or 100 pmol ISD in TAP buffer (50 mM Tris pH 7.5, 100 mM NaCl, 5% (v/v) glycerol, 0.2% (v/v) Nonidet-P40,

1.5 mM MgCl2, and protease inhibitor cocktail (EDTA-free, cOmplete; Roche)) in the presence of 40 U RNase inhibitor (Fermentas) for 60 min at 4 °C on a rotary wheel. Poly(C) (Sigma P9827) or poly(U) (Sigma P8563) agarose beads (20 μl bed volume) were either incubated with excess poly(I) (Sigma P4154) or poly(A) (Sigma P9403), respectively, or left untreated. Beads were washed three times with TAP buffer to remove excess unbound nucleic acids. Cell lysates from mouse RAW macrophages, human THP-1 macrophages, and drosophila S2 cells were prepared by flash-freezing cells in liquid nitrogen, followed by lysis in TAP buffer for 30 min on ice. Whole flies were mixed with TAP buffer and lysed by bead milling using the FastPrep-24 (MPBio) with Lysing Matrix D (MPBio) at 5500 rpm two times for 25 s. Lysates were clarified by centrifugation at 16,000×g for 10 min. Nucleic acid-coated beads were incubated with 2 mg protein of cell lysates for 60 min, washed three times with TAP buffer, and twice with TAP buffer lacking Nonidet-P40 to remove residual detergent. Four independent affinity purifications were performed for each bait. Following affinity purification, bound proteins were denatured by incubation in U/T denaturation buffer (6 M urea, 2 M thiourea, 1 mM DTT (Sigma), 10 mM HEPES, pH 8) for 30 min and alkylated with 5.5 mM iodoacetamide (Sigma) for 20 min. After digestion through the addition of 1 μg LysC (WAKO Chemicals USA) at room temperature for 3 h, the suspension was diluted in 50 mM ammonium bicarbonate buffer (pH 8). The beads were removed by filtration through 96-well multiscreen filter plates (Millipore, MSBVN1210), and the protein solution was digested with 0.5 μg trypsin (Promega) overnight at room temperature. Peptide purification based on C18 Empore filter disks (3 M) was carried out as described previously[75] and peptides were resuspended in buffer A* (0.2% TFA, 2% ACN) for LC–MS/MS analysis.

**TAOK2 affinity purification**. Plasmids coding for truncated (amino acids 1–993) and affinity-tagged (C-terminal V5 tag) wild-type rat TAOK2 or mutated (R702C or D151A) rat TAOK2 variants, or coding for V5-tagged control baits, EGFP and PGAM5, were transfected (PEI) into HEK293T cells. Following induction of bait expression with doxycycline for 1 day, cells were left untreated or stimulated by transfection (PEI) of 5 μg/ml poly(I:C) for 4 h. For each bait and condition, quadruplicate affinity purifications were performed as described previously[75]. Briefly, cell pellets from two 15-cm dishes were lysed in TAP lysis buffer (50 mM Tris-HCl pH 7.5, 100 mM NaCl, 1.5 mM MgCl2, 0.2% (v/v) NP-40, 5% (v/v) glycerol, cOmplete protease inhibitor cocktail (Roche), 0.5% (v/v) 750 U/μl Sm DNase) and sonicated (5 min, 4 °C, 30 s on, 30 s off, low settings; Bioruptor, Diagenode). Following normalization of protein concentrations, cleared lysates were incubated with 20 μl anti-V5-agarose affinity gel (Sigma-Aldrich, A7345) with constant agitation for 3 h at 4 °C. Nonspecifically bound proteins were removed by four subsequent washes with lysis buffer followed by three detergent-removal steps with washing buffer (50 mM Tris-HCl pH 7.5, 100 mM NaCl, 1.5 mM MgCl2, 5% (v/v) glycerol). Enriched proteins were denatured by the addition of SDC lysis buffer (4% SDC, 100 mM Tris-HCl, pH 8.5), followed by reduction and alkylation for 5 min at 45 °C with TCEP (10 mM) and CAA (40 mM) and digested overnight at 37 °C using trypsin (1:100 w/w, enzyme/protein, Sigma-Aldrich) and LysC (1:100 w/w, enzyme/protein, Wako). Peptides were desalted and concentrated using SDB-RPS StageTips (Empore). In brief, samples were diluted with 1% TFA in isopropanol to a final volume of 200 μl and loaded onto StageTips, subsequently washed with 200 μl of 1% TFA in isopropanol and 200 μl 0.2% TFA/2% ACN. Peptides were eluted with 75 μl of 1.25% ammonium hydroxide (NH4OH) in 80% ACN and dried using a SpeedVac centrifuge (Eppendorf, Concentrator Plus). Peptides were resuspended in buffer A* (0.2% TFA, 2% ACN) before LC–MS/MS analysis.

A fraction of the cell pellets used for affinity purification was further prepared for full proteome analysis. To this end, cell pellets were lysed in SDC lysis buffer (4% SDC in 100 mM Tris-HCl, pH 8.5) for 5 min at 9 °C and sonicated (15 min, 4 °C, 30 s on, 30 s off, high settings; Bioruptor, Diagenode). Following normalization of protein concentrations, proteins were reduced and alkylated for 5 min at 45 °C with TCEP (10 mM) and CAA (40 mM), and digested overnight at 37 °C using trypsin (1:100 w/w, enzyme/protein, Sigma-Aldrich) and LysC (1:100 w/w, enzyme/protein, Wako). Peptides were purified with SDB-RPS StageTips as described for TAOK2-AP-MS samples.

**Full proteome analysis of SFV-infected TAOK2 KO THP-1 cells**. For full proteome (FP) analysis, THP-1 cells with KO of TAOK1, -2, -3, STAT1, or NEG4 (control) were mock-treated or infected with SFV. Cell pellets of quadruplicates were lysed (6 M GdmCl, 10 mM TCEP, 40 mM CAA, 100 mM Tris-HCl pH 8), boiled at 99 °C for 10 min and sonicated (15 min, 4 °C, 30 s on, 30 s off, high setting; Bioruptor Plus). Protein concentrations of cleared lysates were normalized to 50 μg and proteins were pre-digested with 1 μg LysC (37 °C, 3 h) followed by a 1:10 dilution (100 mM Tris-HCl pH 8) and overnight digestion with 1 μg trypsin at 30 °C. Peptide purification based on C18 Empore filter disks (3 M) was carried out as described previously[75] and peptides were resuspended in buffer A* (0.2% TFA, 2% ACN) for LC–MS/MS analysis.

**LC–MS/MS measurements**. Purified peptides from nucleic acid affinity purifications (NA-AP), TAOK2 affinity purifications (TAOK2-AP), and full proteome

(FP) samples from TAOK2-AP and TAOK2 KO FP experiments were analyzed by mass spectrometry as described previously[112,113].

Briefly, peptides from NA-AP samples were loaded on a C18 reversed-phase column (15–20 cm Reprosil-Pur 120 C18-AQ, 1.8 μM and 200 mm × 0.075 mm or 3 μM and 150 mm × 0.075 mm; Dr. Maisch) and separated using an EASY-nLC 1200 system (Thermo Fisher Scientific) with a 5 to 30% acetonitrile gradient in 0.5% acetic acid at a flow rate of 250 nl/min over a period of 95 min. The nanoLC system was directly coupled to the electrospray ion source of an LTQ-Orbitrap XL mass spectrometer (Thermo Fisher Scientific) operated in a data-dependent mode with a full scan at a resolution of 60,000 with concomitant isolation and fragmentation of the ten most abundant ions in the linear ion trap.

Peptides from FP and TAOK2-AP samples were loaded on a C18 reversed-phase column (50 cm, 60 °C; 75 μm inner diameter; packed in-house with ReproSil-Pur C18-AQ 1.9 μm silica beads; Dr. Maisch) and separated using an EASY-nLC 1200 system (Thermo Fisher Scientific) with a 5 to 30% acetonitrile gradient in 0.1% formic acid at a flow rate of 300 nl/min over a period of 120 min. Eluting peptides were directly analyzed on a Q-Exactive HF mass spectrometer via a nano-electrospray source (Thermo Fisher Scientific). The data-dependent acquisition included repeating cycles of one MS1 full scan (300–1650 m/z, resolution (R) of 60,000 at m/z 200) at an automatic gain control (AGC) target of $3 \times 10^6$, followed by 15 MS2 scans of the highest abundant isolated and higher-energy collisional dissociation (HCD) fragmented peptide precursors (R of 15,000 at m/z 200). For MS2 scans, the collection of isolated peptide precursors was limited by an AGC target of $1 \times 10^5$ and a maximum injection time of 25 ms. Isolation and fragmentation of the same peptide precursor was eliminated by dynamic exclusion for 20 s. The isolation window of the quadrupole was set to 1.4 m/z and HCD was set to normalized collision energy (NCE) of 27%. All data were acquired in profile mode using positive polarity.

**Bioinformatic analysis of the MS data**. RAW files of NA-AP, FP, and TAOK2-AP datasets were processed with MaxQuant (NA-AP: version 1.5.0.0/ FP + TAOK2-AP:version 1.6.14.0)[114] using the standard settings, label-free quantitation (LFQ), and match between runs enabled. Spectra were searched against forward and reverse sequences of the reviewed human proteome including isoforms (Uniprot KB) as well as SFV (FP) or EGFP (TAOK2-AP) protein sequences by the built-in Andromeda search engine. The MaxQuant output was further analyzed using Perseus (NA-AP: version 1.5.2.1, FP: version 1.6.13.0, TAOK2-AP:version 1.6.15.0), R (version 4.1.0), and R Studio (version 1.4.1717)[115]. Detected protein groups within the protein groups output table identified as known contaminants, reverse sequence matches, only identified by site or quantified in less than three out of four replicates in at least one condition were excluded. LFQ values were log$_2$-transformed and missing values were replaced by sampling values from a normal distribution calculated from the measured data (width = 0.3 s.d., downshift = −1.8 × s.d.).

To identify enriched proteins in the NA-AP dataset, the intensity values in MS runs of NA baits were compared against the controls using a two-sided Welch's *t*-test (S0 = 1; min. 2 valid values in at least one group) with a permutation-based FDR of 0.05 (for the poly(I:C) enrichment in fly an FDR of 0.001 was used). Candidate clustering is based on Euclidian distance and Ward as agglomeration method.

In the FP dataset, differentially expressed protein groups between control (NEG4) and TAOK1, −2, −3, or STAT1 KO THP-1 cells for each treatment were identified via two-sided Student's *t*-test (S0 = 0.1; permutation-based FDR <0.05, 250 randomizations; min. 3 valid values in at least one group).

In the case of the TAOK2-AP dataset, unnormalized LFQ intensities were first normalized by subtraction of the sample-specific median intensity from each protein group intensity and both control baits, EGFP and PGAM5, were combined into one control group for statistical analyses. TAOK2 interacting proteins were identified by comparing each TAOK2 variant against the control group, separately for mock and poly(I:C) stimulated conditions, using a two-sided Student's *t*-test (permutation-based FDR <0.05, 250 randomizations). Significantly enriched proteins were only considered as TAOK2 interactors if they showed a log$_2$ fold change enrichment of ≥1.5. STRING enrichment of wild-type TAOK2 interacting proteins in mock and poly(I:C) stimulated cells was performed in Cytoscape (version 3.8.2) using the stringApp (version 1.6.0) in combination with a confidence cutoff of 0.2 for considering functional connections and an MCL inflation parameter of 4 for clustering.

Protein superfamily domain annotations were identified using the CDD batch search[116]. IAV-regulating proteins were described, all proteins confirmed in one or more screens were considered[41]. For overlaps to NA-interaction data, candidates that were identified in at least one screen were considered. Overlaps with proteins changing RNA interaction in SINV infected cells were obtained from Table S1 "18 hpi quantitative" in Garcia-Moreno et al.[17]. Significant enrichment was calculated via Fisher exact test (Benjamini–Hochberg adjusted FDR <0.05). A list of known NA binders was taken from Binns et al.[117] and compared to the identified proteins to determine the percentage of known NA binders. For AP EnrichmentMap[118] (version 2018.12) was used to annotate the human proteins with Gene Ontology (GO) terms. To identify the terms that are specifically enriched among the protein binders of specific NA bait or shared by multiple NA baits, the OptEnrichedSetCover.jl Julia package was used (https://github.com/alyst/

OptEnrichedSetCover.jl). FP annotation with Gene Ontology terms corresponding to biological processes (GOBP), molecular functions (GOMF), and cellular compartments (GOCC) was performed within Perseus (downloaded from http://annotations.perseus-framework.org, 06.2019). Testing for the enrichment of annotations within significantly changing protein groups was done using Fisher exact test with the Benjamini–Hochberg adjusted FDR cutoff set to 0.05. Orthologues mapping was performed using DRSC Integrative Ortholog Prediction Tool[51], excluding the orthologous with a DIOPT score less than 2 (low score). Identified human orthologues were filtered against an experimentally determined THP-1 proteome, so as to not include orthologues of proteins that could not be experimentally identified in THP-1 cells in the interspecies comparison. Upstream promoter analysis was performed using iRegulon[68].

**Selection of Candidates**. After the statistical analysis of the data, a score was calculated for each bait-control comparison (1):

$$(\text{foldchange} \cdot -\log(\text{pvalue}))^5 \cdot \text{proteome abundance}^{0.05} \cdot$$
$$\text{number of affinity purifications with valid values}^{0.01}. \tag{1}$$

For each bait, the top 200 protein candidates of humans were compared to the 200 best mouse ones. Final candidates were selected factoring in the regulation of potential candidates by type-I or type-II interferon (fold change >2) (interferome.org) and excluding known nucleic acid sensors and proteins involved in transcription. For the drosophila candidate list, the 10% of interactors with the highest score were compared to the final human/mouse candidate list and fly proteins with orthologues were selected. The remaining proteins of the fly candidate list were selected based on the knowledge available from literature and predicted GO terms.

**CRISPR/Cas9 KO screen in human cells**. sgRNA sequences targeting 90 candidate genes and the positive control STAT1 were selected using the GPP sgRNA designer[119] (Supplementary Data 10). The sgRNA sequences (three per gene), including 12 negative controls previously tested in human cells[120], were cloned into lentiCRISPRv2 vector (Addgene # 52961), carrying the *Streptococcus pyogenes* Cas9 enzyme and puromycin resistance. Lentiviral particles were generated by transient transfection of sub-confluent HEK293T cells (ATCC, grown in DMEM supplemented with 10% FCS and 1% penicillin/streptomycin (Gibco) with the lenti-CRISPRv2, psPAX2 (Addgene #12260), and pMD2.G (Addgene #12259) vectors, using PolyFect (Qiagen). The media was exchanged to RPMI supplemented with 10% FCS and 1% penicillin/streptomycin (Gibco) 24 h post-transfection. Viral supernatants were collected 72 h post-transfection, filtered, and stored at −80 °C until further use. THP-1 cells were seeded at 5E5 cells/mL in 400 and 600 μL of lentivirus (200 μL per sgRN9) were added. After overnight incubating the medium was replaced with 1 μg/mL puromycin (Sigma P8833) containing medium. The cells were maintained and scaled up in 1 μg/mL puromycin-containing medium for 16 days. For the viral screening, the cells were seeded at 0.5E5 cells/well in a 96-well plate, each cell line was seeded in technical triplicates. Half of the THP-1 cells were differentiated with PMA upon seeding overnight before stimulation. One day after seeding, the cells were infected with one of four viruses; HSV-1-Firefly Luc (MOI 0.2), Influenza A SC35M NS1_2A_Gaussia_2A_NEP (MOI 0.1), SFV6-2SG-Gaussia-Luc (MOI 0.1), and VSV-Firefly Luc (MOI 0.1). Seventeen hours after infection cell viability, as well as the accumulated luciferase signal, were measured. To determine cell viability, 50 μg/mL of resazurin (Sigma R7017) were added to each well and incubated at 37 °C for 30 min, after which the fluorescence (535/590 nm) was measured using an Infinite 200 PRO series microplate reader (Tecan). Gaussia luciferase levels were determined; 20 μL of cell supernatant was mixed with 20 μL of Gaussia Luciferase buffer (20 mM MOPS, 75 mM KBr, 1 mM EDTA, 5 mM MgCl2, pH adjusted to 7.8 with 300:1 of a coelenterazine (Carl Roth #4094.3) solution 3 mM in acidified methanol) and incubated at RT in the dark for 5 min, after which the luminescence was measured using an Infinite 200 PRO series microplate reader (Tecan). The levels of firefly luciferase were measured by first pelleting cells (800 rpm for 5 min), followed by resuspension in 50 μl 1x Passive lysis buffer (Promega #E1941) and a freeze-thaw cycle to break the cell membrane. About 20 μl of cell lysate was mixed with 20 μl of firefly substrate (20 mM Tricine, 3.74 mM MgSO4, 33.3 mM DTT, 0.1 mM EDTA, 270 μM Coenzyme A trilithium salt (Sigma-Aldrich #C3019), 470 μM D-Luciferin sodium salt (Sigma-Aldrich #L6882), 530 μM ATP disodium salt (Sigma-Aldrich #A7699), pH 7.8-8), and incubated at RT in the dark for 5 min, after which the luminescence was measured using a plate reader. Statistical analysis of the luminescence data was done in R (v3.3). Cell viability and luciferase data were fit using the random-effects generalized linear Bayesian model, which, in R glm notation, could be expressed as (2):

$$log_2(\text{intensity}) \sim 1 + \text{batch} + \text{virus} * \text{gene} \tag{2}$$

The effects corresponding to the screen batch, the virus infection (virus), gene KO (gene), and the effect of interaction between the last two model factors were set to have horseshoe prior distribution[121]. The distribution of log$_2$(intensity) was set to be Laplacian for robust handling of outliers. The model was fit with the No-U-Turn Markov Chain Monte Carlo sampler implemented in rstan R package (ver. 2.15[122]). About 2000 iterations of the sampling method (1000 warmup + 1000 sampling) in eight independent MCMC chains were done. The

model parameters samples were collected at each second iteration of the MCMC run. To estimate the significance of the viral replication change/cell viability, the reconstructed batch effect-free posterior distribution of luciferase intensity upon virus infection and gene KO ($Luc_{KO}$) was compared with the posterior distribution of NT control ($Luc_{NT}$). The significance was defined as the probability that the $\log_2$ fold change of luciferase intensity is different from zero (3):

$$P_{value} = 2 \cdot \min(P(\log_2(Luc_{KO}/Luc_{NT}) < 0), P(\log_2(Luc_{KO}/Luc_{NT}) > 0)). \quad (3)$$

No $P$ value correction for multiple hypothesis testing was done since this is handled by the choice of model parameters prior to distribution.

**Knockdown screen in flies**. KK and GD inverted repeat transgenic fly lines from the VDRC stock center were used to induce the knockdown of candidate genes (Supplementary Data 10). shmCherry (BDSC #35787) and shAGO2 (BDSC #34799) were used as controls. Transgenic males containing shRNA and the inverted repeat of the target gene under the control of Gal4 regulated upstream activating sequence (UAS) were crossed with virgin females [Actin-Gal4/CyO; Tubulin-Gal80$^{TS}$] at 18 °C. The F1 generation confirming genotype was placed at 29 °C for 5–7 days to induce the knockdown of candidate genes. All experiments were subsequently done at 29 °C.

Viral stocks were prepared in 10 mM Tris-HCl, pH 7.5. Infections were performed with 6–8 days old adult flies by intrathoracic injection (Nanoject II apparatus, Drummond Scientific) with 4.6 nL of viral particle solution (500 pfu/fly for DCV and FHV, 5 pfu/fly for CrPV, 2500 pfu/fly for SINV, 10,000 pfu/fly for VSV). Injection of the same volume of 10 mM Tris-HCl, pH 7.5, was used as a control. Infected flies were frozen, three males and three females per condition, for RNA isolation at the indicated time points.

Total RNA from flies was isolated using a NucleoSpin 96 kit or manually using Trizol Reagent RT bromoanisole solution (MRC), according to the manufacturer's instructions. One microgram of total RNA was reverse transcribed using an iScriptTM cDNA synthesis kit (Bio-rad). About 100 ng of cDNA was used for quantitative real-time PCR (RT-qPCR), using iQTM Custom SYBR Green Supermix Kit (Bio-rad) for fly samples, according to the manufacturer's instructions, on a CFX384 Touch Real-Time PCR platform (Bio-Rad). Primers targeting viral sequences are listed in Goto et al. and in Supplementary Table 1[123].

All statistical analysis was done in R (version 3.5.0). ΔCq was calculated by subtracting CqVirus from CqRP49. Significance was calculated using lsmeans R package (version 2.30) with Dunnet's adjustment for p values.

**Recombinant dTao kinase activity and affinity measurements**. The recombinant full-length dTao (CG14217) was produced by the Core Facility of the Max Planck Institute of Biochemistry. dTao was cloned into pCoofy27 (Addgene #44003) for baculovirus-based expression in High Five cells[124]. Cells were lysed via douncing (1 mM AEBSF-HCl, 2 μg/mL Aprotinin, 1 μg/mL Leupeptin, 1 μg/mL Pepstatin, 2,4 U/mL Benzonase, 2 mM MgCl2). Protein purification was performed using the coupled N-His6 tag via affinity purification (Ni Sepharose High-Performance GE) in His Binding Buffer (50 mM Na-P, 500 mM NaCl, 10 mM Imidazole, 10% Glycerin, and 1 mM TCEP, pH 8) at 4 °C for 2.5 h and washed with His Wash Buffer (50 mM Na-P, 500 mM NaCl, 20 mM Imidazole, 10% Glycerin, and 1 mM TCEP, pH 8). Purified protein was eluted from the beads using His Elution Buffer (50 mM Na-P, 500 mM NaCl, 250 mM Imidazole, 10% Glycerin, and 1 mM TCEP, pH 8). The protein was further purified by gel filtration (HiLoad 26/60 Superdex 200 GE) and eluted in Storage Buffer (20 mM Tris, 200 mM NaCl, 10% Glycerin, 0.2 mM EGTA, and 1 mM TCEP, pH 8) and concentrated (Amicon Ultra 15) at 3700 rpm, 4 °C in 5 min steps. The production was verified by LC–MS. The kinase activity in the presence or absence of poly(I:C) was determined using the ADP-Glo™ Kinase Assay kit (Promega; V9101) and performed according to the manufacturer's instructions. Briefly, dTao was mixed with a substrate, ATP, and poly(I:C) or additional buffer, and incubated. After the incubation, the remaining ATP was depleted, ADP was converted to ATP which was then used for a luciferase reaction.

To determine the affinity between dTao and poly(I:C) a fluorescence quenching assay was used. Briefly, the fluorescence of the FITC-tagged poly(I:C) (2.5 μg/mL) was measured in presence of increasing concentrations of dTao or denatured dTao (2% SDS (Sigma) boiling at 95 °C for 5 min). The analysis was performed using the built-software Affinity Analysis v2.2.4 (NanoTemper Technologies MO) for Initial fluorescence.

**Live-cell imaging, RT-qPCR, and cytokine measurements**. THP-1 cells CRISPR/Cas9 targeted for the indicated gene were seeded at 2.5E5 cells/mL. After overnight incubation, the cells were infected with the indicated fluorescent viruses and the fluorescent signal was followed over time. A549-IFIT1-eGFP CRISPR/Cas9 targeted for the indicated gene were seeded at 5E3 cells/mL. After overnight incubation the cells were infected with the indicated viruses, treated with IFN-α B/D (1000 units/mL) or transfected with poly(I:C) (2 μg/mL), 100 ng/mL in vitro transcribed triphosphorylated hairpin RNA (IVT4)[125], 100 ng pTO-SII-HA-MAVS expression plasmid or PBS as a control using METAFECTENE Pro® (2 μg/mL, Biontex), and the fluorescent signal was followed over time. A549-ACE2 cells were

seeded at around 50% confluence, incubated overnight, and pretreated for 6 h with RAF265 (10 μM) before infection with GFP-expressing SARS-CoV-2 reporter virus (MOI 3). Fluorescence intensity was measured every 2–4 h using an Incucyte S3 fluorescence light microscopy screening platform (Sartorius). The fluorescence intensity of the reporter was assessed as integrated intensity per image normalized on cell confluence per well using IncuCyte S3 Software (Essen Bioscience; version 2019B Rev2). Two-way ANOVA with Geissser–Greenhouse correction and Sidak's multiple comparisons test was performed with GraphPad Prism (version 9.1.0) to evaluate the significance.

Total cellular RNA was harvested and isolated using MACHEREY-NAGEL NucleoSpin RNA mini kit according to manufacturer instructions. Reverse transcription was performed using Takara PrimeScript RT reagent kit with gDNA eraser according to manufacturer instructions. RT-qPCR was performed using primers targeting GAPDH (for: GATTCCACCCATGGCAAATTC; rev: AGCATCGCCCCACTTGATT), SFV (for: GCAAGAGGCAAACGAACAGA; rev: GGGAAAAGATGAGCAAACCA), and MX1 (for: TGGAGGCACTGTCAGGA GTT; rev: CCACAGCCACTCTGGTTATG). PowerUp SYBR Green (Thermo Fisher, A25778) was used on QuantStudio 3 Real-Time PCR system (Thermo Fisher). Ct values, obtained using QuantStudio Design and Analysis Software v1.4.3, were averaged across technical replicates and fold change values were used as a measure of gene expression (calculated from ΔΔCt method, calibrated to control or mock control for SFV and MX1, respectively).

The ELISA and cytometric bead array were used according to the manufacturer's protocol and measured using an Infinite 200 PRO series microplate reader (Tecan) and Bio-Plex 200 Luminex Technology, respectively.

**Reporting Summary**. Further information on research design is available in the Nature Research Reporting Summary linked to this article.

## Data availability

The mass spectrometry proteomics data have been deposited to the ProteomeXchange Consortium (http://proteomecentral.proteomexchange.org) via the PRIDE partner repository with the dataset identifier PXD027894, PXD027896, and PXD027919. Source data are provided with this paper.

## Code availability

Pathway enrichment analysis was performed with the OptEnrichedSetCover.jl Julia package (https://github.com/alyst/OptEnrichedSetCover.jl).

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

## Acknowledgements

We want to acknowledge the immunopathology of virus infections laboratory for critical discussions and suggestions. We further thank Korbinian Mayr, Igor Paron, and Gaby Sowa for maintaining mass spectrometers and the MPI-B core facility, especially Judith Scholz, Leopold Urich, Sabine Suppmann, and Stephan Uebel, for support, as well as Silke Hegenbarth and Percy Knolle for the use of the Bio-Plex 200 Luminex Technology. We are grateful to Bernhard Küster and Jonathan Morris for sharing their experimental data and protocols, regarding RAF265. We further thank Ralf Bartenschlager for IFIT1-eGFP A549 cells, Soren Riis Paludan for HSV-1-Luc, Peter Reuther for IAV-Luc, Gert Zimmer for VSV-Luc, Volker Thiel for SARS-CoV2-2-GFP, Andres Merits for SFV-NanoLuc, SFV-mCherry, and antibodies, Bob Silverman for RNase L antibody, and Melanie Cobb for the TAOK2 expression plasmid. We also thank Simon Giosele and Swayanka Biswas for their assistance. We would like to thank Emilie Lauret and Stefanie Pietsch for their excellent technical assistance during the fly screen. We would like to thank Giuseppe Fiume and Justyna Konecka for their assistance in generating the CRISPR/Cas9 plasmid library. Work in the author's laboratories was supported by the Max Planck Free Floater program, an ERC consolidator grant (ERC-CoG ProDAP,

817798), the German Research Foundation (PI 1084/3, PI 1084/4, and PI 1084/5 and TRR179, TRR237, and MS 632/15) to A.P., Infect-ERA, ANR, and the German Federal Ministry of Education and Research (COVINET) to A.P. and (ERASE) to J.-L.I., G.S.-F., and A.P. the Labex NetRNA (ANR-10-LABX-0036_NETRNA) and the Equipex I2MC to J.-L.I. and C.M.

## Author contributions

F.L.P., A.M., C.U., M.H., C.H., Y.B., L.L.A., L.O., T.M.L. and V.G. conducted experiments. F.L.P., C.U., A.S., M.H. and R.E.B. analyzed data. E.G., G.S.-F. and R.H. contributed critical reagents. F.L.P., A.M, C.U., M.H., C.M., J.-L.I. and A.P. designed the experiments and wrote the paper.

## Funding

## Competing interests

The authors declare no competing interests.
