## [Peer Review File · Nature Communications]

Cross-species analysis of viral nucleic acid interactors identifies TAOs as immune regulatorsREVIEWER COMMENTS

Reviewer #1 (Remarks to the Author):

In an effort to reveal new factors implicated in antiviral defenses, the author have identified evolutionary-conserved proteins that associates with nucleic acids. They combined mass-spectrometry with targeted validation in viral infectivity assays. Several new candidate factors have been identified. For mechanistic validation of one candidate, they picked TAOK, a family of kinases previously unrelated to viral defenses. They show that TAOK binds directly to dsRNA, that dsRNA modulates kinase activity and that TAOK is required to mount a full interferon response against RNA virus infection. Overall, this is an impressive study that constitutes a highly valuable resources to mine for novel factors. The validation of TAOK, although mechanistically limited, confirms the power of this approach and provides a first exciting validated hit that will undoubtedly lead to future studies.

Overall, the manuscript if very well written and the figures are clear. The data is convincing and I have no major issue with this study. Nonetheless, the current manuscript should be improved for publication. Furthermore, while in-depth mechanistic analysis of TAOK is not expected, a few key experiments appear missing to fully demonstrate the relevance of this factor.

Main comments:

1) The data supporting of TAOK is convincing, however, a few more experiments are missing to fully grasp the significance of this family of kinases. While a deep mechanistic analysis is not expected, key points are somewhat confusing or unclear and should be improved with relatively simple experiments:

- * Fig 6g/h: While there is no doubt that RAF265 and Comp43 may inhibit TAOK, the data currently does not exclude that the effects are due to targeting of other proteins. The specificity should be assessed by comparing the effect on TAOK KO vs WT cells.

- * TOAK1, 2 and 3 all appear to independently contribute to the ISG response (Fig 5d). However, the degree of overlap between the affected ISG responses is not clear (Fig 5d). What is the relative contribution of the 3 TAOKs to the induction of IFN? Of ISGs? To the induction of the antiviral state? A definitive experiment would be to compare single vs triple KO cells in the IFN induction, ISG induction and the antiviral activity.

- * The specificity of TAOK to NAs is lacking and there is currently no evidence linking the functional effect on IFN/ISG/infection to RNA. They have access to a nice panel of various NAs. What is the IFN and ISG responses of WT vs TAOK KO cells to the various RNA and DNA ligands?

2) They highlight the similarity between OAS and cGAS (Ref 6). They identify that ABCF1 associated with 2'5'OA, a product of OAS (Fig S1b). Considering the effect of ABCF1 depletion on HSV replication, a cGAS-sensitive virus, does ABCF1 associate with cGAMP, the product of cGAS? How does it compare to 2'5'OA?

3) Fig 5c, Fig 6, Fig 7: The statistical analysis and reproducibility description for mechanistic experiments should be improved:

- * indicate what lines and errors bar represent

- * Fig 5c, Fig 7: conventional symbols for statistical significance should be shown on the graphs

- * The t-test should not be used on datasets that contain $n > 2$ groups on the graph. They should use a one-way ANOVA or a non-parametric Friedman test, for example.

- * Fig 7: the cytokine values are very 'tight' across the 4 replicates, which is highly unusual on linear scales (cytokines can show decades of variations and are often shown on log scales). Are these 'biological replicates' from independent experiments, or combined from technical replicate wells within experiments? How many times the experiments have been actually performed?

- * Fig 7e: I don't think G-CSF is induced at all here. They should use statistics to evaluate which cytokines are actually induced and to be considered valid.

Other comments:

4) There is a confusion between "nucleic acid binding" and "proteins that associate with nucleic

acids" (e.g. line 2-4 of abstract, 1st paragraph of page 6). For many readers, "binding" implies a direct interaction. The current mass spec approach identifies association or partners, but not direct binding. Few candidates are evaluated by IP (Fig S1b), but again this does not test direct interaction. Direct binding is only demonstrated for dTao (Fig 5b), which is very nice. They should carefully reformulate the text to avoid the confusion between binding and association.

5) Fig S1a, S3, S6, 2: Heatmaps show 3 columns per condition but this is not explained in the legend. Also, it could help to use different text colors for the labels of test NAs vs control NAs.

6) Fig 1b: CDKN2AIP is mentioned in the text but I do not see it in the figure.

7) Fig 1c: I don't understand the categories shown.

8) Page 7: They highlight MATR3, SFPQ, PSPC1, NONO and RBM14 in association with the RNA NAs. These proteins are all members of the paraspeckle complex (e.g. PMID 28977530, 29289458) that contains RNA. However, they refer instead to the HDP-RNP, proposed in one study, which was defined by the combined associated of paraspeckle proteins with proteins involved in the DNA damage response (e.g. DNA-PK) and DNA sensing (cGAS, STING). As currently stated, they detect paraspeckles proteins co-associated with RNA NAs, but this is not an evidence for the hypothetical HDP-RNP. Is there a significant association of DNA-related HDP-RNP proteins found with the paraspeckles proteins, for given NAs? In other words, are they detecting paraspeckles, or HDP-RNP?

9) Figure S3: It is surprising that no protein appears significantly associated with the DNA ligands. There are proteins listed in Table S4/S5 that associated with dsDNA. Why exclude the DNA-binding proteins from Figure S3? Little is known about DNA sensing in fly, so it would be interesting to read their opinion on this in the discussion as well, based on their results.

10) Fig 3b, age 12: In another recent study, UHRF1 was shown to suppress the antiviral IFN response (PMID 33616624). The conflicting results should be mentioned and perhaps discussed.

11) Fig 3b, 4b: p-val circles on red cells are not visible in print. Please use a different color scheme.

12) They identify DDX41 as a clear partner of RNA ligands, but not of DNA ligands. Considering the controversy in the literature on the role of DDX41 in DNA sensing, do they foresee a role of RNAs? Can they discuss this?

Reviewer #2 (Remarks to the Author):

The authors report a very nice method of unbiased screening for conserved antiviral proteins using mass spectrometry in three species: fly, mouse, and human to detect proteins binding to nucleic acids common to a diverse set of viruses. The overlapping candidates were then functionally probed for their ability to modulate viral replication by an in-vitro CRISPR-cell based screen and in vivo knockdown of targets in flies. One of the best candidates, Tao kinase was shown to have specific binding to Poly I:C. The effect of Tao kinases was also tested with some putative inhibitors of TaoK, which enhanced viral replication. TAOK's contribution to signaling was narrowed down to IRF3 but not NF- κ B mediated signaling. Several new targets for investigation of anti-viral proteins have been described and warrants further follow up or may provide corroborating evidence for other current studies. There are some important changes/ additions needed to enhance the conclusions of the report.

Major critiques:

1. The role of TAOK2 is convincingly shown, but to complete the mechanistic connection to known antiviral pathways, I would suggest the authors pulldown TAOK2 and perform Mass Spectrometry

to identify binding partners (IP-MS). This could be performed across species or in a single species with confirmation by Western Blot. Given the author's expertise in MS assays and cell manipulations, this should be an achievable experiment that will provide convincing evidence of the role of TAOK2.

2. The data in Figure 4 are currently organized alphabetically, but I would suggest they be organized based on the nucleic acid type they bound in the initial screen. This would allow for a direct comparison of how they impacted viral replication. Ex: proteins binding to DNA vs. the dsDNA virus HSV1.

3. Statistical comments: For figures 3b and 4b, I suggest changing $-\log_{10}$ p-value from 10 to 0.01 etc. It was confusing to interpret without reading a few times.

a. For Figure 6, and 7, please include p-values between matched conditions.

Minor comments:

1. The 2nd sentence and second to last sentence of the first introductory paragraph and vague and specific cytokines and cellular machineries should be named or re-worded. Additionally "specific marks" line 5 paragraph 2 of introduction should be more clear.

2. Italicize gene names where appropriate (human genes).

3. Spell out AP-MS the first time the abbreviation is used.

4. Pg 5 line 4, please provide a reference for the NA binding properties of IFIT1, RIG-I and DHX9.

5. SMARCA5 had some of the strongest viral replication present when knocked out (Fig. 3B). Why was it not followed up on for investigation?

6. Page 20: The evidence provided suggests, but does not "indicate" TAOK evolved in parallel to TLR and cGAS- more advanced phylogeny and genetic studies would be needed for this conclusion. Also, what is the point of the next sentence after this. It appears as non-sequeter and does not summarize the preceding paragraph nor transition to the next. I suggest moving or removing.

7. Though not required for this study, knocking out TAOK2 or other targets in mice and performing comprehensive immune analysis would be a fruitful future direction.

Signed,
Jeff SoRelle, MD

Reviewer #3 (Remarks to the Author):

In the manuscript by Pennemann et al., the authors describe a screen for host nucleic acid sensors using human, mouse and fly model systems. The immense amount of data generated through this screen is analyzed well, and experiments to validate the hits uncovered in the screen are clearly performed using the relevant model systems. The authors' tour de force in the screen and validation is highly commendable. Following the experimental validation of the protein candidates, the authors focus on the TAO kinases for functional assays. This section needs further development to justify the authors conclusions, e.g., a statement in the abstract regarding TAOK2 and regulation of IRF3 versus NFkB. Also, this section mostly focuses on the mammalian TAOs even though dTao was quantitatively analyzed for dsRNA binding and activation. Below are suggestions to the authors that would improve their manuscript.

MAJOR:

A) Figure 4 should include a viral pathogen with a dsDNA genome, similar to the validation experiment with HSV-1 and the mammalian CRISPR screen.

B) Figure 5/Page 20 – the authors state that TAOs and dTao bind dsRNA with high affinity. Without a proper control for the experiments in Figure 5A-C, the conclusion is not fully justified. The authors should include known mammalian and fly dsRNA interactors in these experiments.

- C) Even though the authors provide quantitative data showing that dTao interacts with and is activated by dsRNA, no further functional validation is performed in the fly model. This is surprising considering the number of reagents available to evaluate dTao in a fly model. These experiments are needed to support the role of dTao in the fly.
- D) RAF265 and Compound 43 function on the host responses, both in the context of control and TAOK2 KO, should be examined, as described in Koo et al.
- E) Page 17 – the authors conclude that TAOK2 affects IRF3 but not NFkB (also stated in the abstract). However, the data do not fully support this conclusion. In the same paragraph they state that MCP-1 is induced by NFkB, and the data show that MCP-1 secretion is indeed affected by TAOK2. Nevertheless, the authors should perform assays to directly test IRF3 and NFkB activation during SFV infection and TAOK2 KO to properly support their conclusions.

MINOR:

- A) In the abstract, the authors state 182 candidates, 98 restriction factors, and 16 proteins, but these numbers do not agree with what is presented in Figure 8. Further, in Figure 8, are the 28 proteins shown in panel B the 28 at the intersection of the “Candidates” Venn diagram? Which are the 14 that validated by functional screening?
- B) The introduction section should briefly describe the results from the functional assays on TAOK shown in Figures 5-7.
- C) Page 13 – dTao should be discussed in the fly functional screen section.
- D) Figure 7 – statistical tests should be shown for each figure panel.
- E) Discussion – the authors describe the importance of cGAS at the beginning of this section, but cGAS was not found in their screen. Can they speculate about why cGAS was absent from the initial screening results?

Reply to the Reviewers' comments

We want to thank the reviewers for their positive and very constructive comments, which we feel further improved the manuscript. Below you find answers to the individual points.

With best regards,

Andreas Pichlmair and the authors of this manuscript

Reviewer #1 (Remarks to the Author):

In an effort to reveal new factors implicated in antiviral defenses, the author have identified evolutionary-conserved proteins that associates with nucleic acids. They combined mass-spectrometry with targeted validation in viral infectivity assays. Several new candidate factors have been identified. For mechanistic validation of one candidate, they picked TAOK, a family of kinases previously unrelated to viral defenses. They show that TAOK binds directly to dsRNA, that dsRNA modulates kinase activity and that TAOK is required to mount a full interferon response against RNA virus infection. Overall, this is an impressive study that constitutes a highly valuable resources to mine for novel factors. The validation of TAOK, although mechanistically limited, confirms the power of this approach and provides a first exciting validated hit that will undoubtedly lead to future studies.

Overall, the manuscript is very well written and the figures are clear. The data is convincing and I have no major issue with this study. Nonetheless, the current manuscript should be improved for publication. Furthermore, while in-depth mechanistic analysis of TAOK is not expected, a few key experiments appear missing to fully demonstrate the relevance of this factor.

Main comments:

1) The data supporting of TAOK is convincing, however, a few more experiments are missing to fully grasp the significance of this family of kinases. While a deep mechanistic analysis is not expected, key points are somewhat confusing or unclear and should be improved with relatively simple experiments:

** Fig 6g/h: While there is no doubt that RAF265 and Comp43 may inhibit TAOK, the data currently does not exclude that the effects are due to targeting of other proteins. The specificity should be assessed by comparing the effect on TAOK KO vs WT cells.*

This is a valuable request to further characterize the activity of the kinase inhibitors. The only way to unambiguously show that the inhibitors are selectively targeting TAO kinases would be to test the activity of the inhibitors in TAOK1, -2, -3 triple knockout cells. The assumption would be that the inhibitors are less active in these cells. However, despite intensive attempts we were unable to generate A549 and THP-1 triple knockout cells lacking TAOK-1, -2 and -3 (Figure R1). We therefore cannot perform the requested experiment for technical reasons. However, we treated control and TAOK2 knockout A549-IFIT1-GFP cells with RAF265 and monitored the expression of IFIT1-GFP in response to SFV. We found that the activity of RAF265 was dramatically reduced in the TAOK2 KO as compared to the activity of the Inhibitor in wt cells. We feel that this assay nicely demonstrates that RAF265 inhibits TAOK2 activity. Please note that we took out data on Compound 43 since we had only used this inhibitor for a single experiment. The new data on RAF265 specificity is shown in Supplementary Figure 7f.

** TOAK1, 2 and 3 all appear to independently contribute to the ISG response (Fig 5d). However, the degree of overlap between the affected ISG responses is not clear (Fig 5d). What is the relative contribution of the 3 TAOKs to the induction of IFN? Of ISGs? To the induction of the antiviral state? A definitive experiment would be to compare single vs triple KO cells in the IFN induction, ISG induction and the antiviral activity.*

This would indeed be very interesting to test. Unfortunately, as mentioned in the previous point, we were unable to generate the required cell lines to test the effect all TAOKs. However, to further test the involvement of TAO kinases we used the individual TAOK1, -2 and -3 deficient cells to confirm and better quantify the mass spectrometry based proteomics data by western blotting and qRT-PCR. Clearly, the individual TAO kinases contribute to induction of the type-I interferon response and to the expression of interferon stimulated genes. We also compared the response to STAT1 deficient cells, which are fully devoid of interferon signaling and serve a positive control. In line with the mass spectrometry data we find that all TAO kinases are individually contributing to antiviral responses and to expression of the interferon stimulated gene MXA. However, the individual depletion of TAO kinases does not lead to a complete reduction of IFN signaling, likely due to a degree of redundancy of individual TAO kinases. This additional data is now provided in Figure 5g, h and Supplementary Figure 7g, h and underline the functions of TAOK1, -2 and -3. We agree with reviewer 1 that additional work on TAO kinases will be important to further understand the activity and contribution.

** The specificity of TAOK to NAs is lacking and there is currently no evidence linking the functional effect on IFN/ISG/infection to RNA. They have access to a nice panel of various*

NAs. What is the IFN and ISG responses of WT vs TAOK KO cells to the various RNA and DNA ligands?

The specificity of TAOKs is suggested from their affinity towards poly-I:C (Fig. 2, Fig. 5a-c). We now address the specificity of TAOK2 for poly-I:C in Supplementary Fig. 7e. A549-IFIT1-GFP wt and TAOK2 knockout reporter cells were transfected with IVT4, which is short RIG-I stimulating PPP-RNA, and with transfection of MAVS, a molecule downstream of RLRs. Both stimuli induced similar amounts of IFIT1-GFP in wt and TAOK2 knockout cells, while poly-I:C and dsRNA generating SFV were more effective in wt as oppose to TAOK2 deficient cells (Fig 6c-e). We also tested DNA transfection in the same cells but due to the known absence of operative DNA sensing in A549s neither of the cell lines induced detectable IFIT1-GFP signal (data not shown). We feel that this experiment underlines the specificity of TAOK2 for dsRNA induced antiviral responses.

2) They highlight the similarity between OAS and cGAS (Ref 6). They identify that ABCF1 associated with 2'5'OA, a product of OAS (Fig S1b). Considering the effect of ABCF1 depletion on HSV replication, a cGAS-sensitive virus, does ABCF1 associate with cGAMP, the product of cGAS? How does it compare to 2'5'OA?

We thank reviewer 1 for this thoughtful comment. We here address this notion on different levels:

1. We characterized the binding of ABCF proteins to 2'5'OA and found that this binding critically requires 5'terminal phosphates (Supplementary Fig. 1b). Dephosphorylated 2'5'OAs do not allow binding of ABCF1 to 2'5'OA. The cyclic nature of cGAMP would not allow presentation of terminal phosphates and therefore binding of cGAMP to ABCF1 and -3 is unlikely.
2. Still, to experimentally address whether cGAMP associates to 2'5'OA or other cellular proteins we employed the mass spectrometry pipeline described in the present manuscript. We used bone-marrow derived macrophages (BMDMs), which we validated to be STING-signalling competent (data not shown). However, in our hands, the available reagent (biotinylated 2'3'cGAMP) was not suitable to significantly enrich for any protein, not even STING, which should associate with cGAMP (Figure R1). The crystal structure of STING bound to

Figure R1: biotin-cGAMP vs cGAMP were added to lysates from BMDM for 1h and used for precipitation with Streptavidin beads followed by LC-MS/MS analysis. Volcano blot show average fold change abundance (x-axis) and corresponding p-values (y-axis) of three independent precipitates.

cGAMP suggests that this inability to precipitate STING is likely due to a steric hindrance of the affinity tag (biotin) in the STING binding groove. Indeed functional experiments also indicate that the biotinylated reagent is not suitable to induce type-I interferons (own experience and personal communication with Veit Hornung, LMU Munich). Therefore, this experimental approach is unfortunately not suitable to address the question.

3. We extensively mined literature for potential cGAMP interactors. An unbiased mass spectrometry approach suitable to test for protein-nucleic acid interactions and therefore likely to identify ABCF1/-3 – cGAMP associations and that is not relying on affinity enrichment is thermal-stability profiling followed by LC-MS/MS. This assay is based on the change in the thermostability of proteins upon ligand binding. Huber et al. (Nature Methods. 2015 Nov;12(11):1055-7) found that STING showed a difference in thermal stability when cGAMP was provided to cells. However, there is no evidence that ABCF1 or ABCF3 are affected by cGAMP treatment, further indicating that these proteins are not associating to cGAMP.

Collectively, we do not see evidence for ABCF1 or -3 binding to cGAMP. We describe the requirement for phosphorylated 2'5'OA in relation to Supplementary Figure 1b but would like to avoid eluding on cGAMP in the same section in order not to perturb the flow of the manuscript.

3) Fig 5c, Fig 6, Fig 7: The statistical analysis and reproducibility description for mechanistic experiments should be improved:

** indicate what lines and errors bar represent*

** Fig 5c, Fig 7: conventional symbols for statistical significance should be shown on the graphs*

** The t-test should not be used on datasets that contain $n > 2$ groups on the graph. They should use a one-way ANOVA or a non-parametric Friedman test, for example.*

We thank the reviewer for his/her comments regarding the statistical analyses since the clarity of data representation is indeed an important aspect. We applied a two-way ANOVA with Šídák's multiple comparisons testing for Figure 5c and Figure 7 to identify statistically significant changes upon dTao binding to poly(I:C) or induction of cytokines upon SFV infection and TAOK2 knockout, respectively. To determine statistically significant changes in case of the live-cell imaging data presented in Figure 6, we used a repeated measurements two-way ANOVA analysis with Šídák's multiple comparisons testing, which was performed to compare treatment versus control conditions. Lines and error bars in these figures represent the mean \pm s.d. and statistical annotations were generalized as follows: ns, p-value > 0.05 ; *, p-value < 0.05 ; **, p-value < 0.01 ; ***, p-value < 0.001 ; ****, p-value < 0.0001 . These information is now provided in the figure legends.

** Fig 7: the cytokine values are very 'tight' across the 4 replicates, which is highly unusual on linear scales (cytokines can show decades of variations and are often shown on log scales). Are these 'biological replicates' from independent experiments, or combined from technical replicate wells within experiments? How many times the experiments have been actually performed?*

The data in question was generated from individually treated wells that were analysed at the same time. Every single data point is thus representing an individual stimulation experiment. The beauty of the cytometric bead array is that the individual cytokines can be measured in parallel in the same reaction, therefore allowing simultaneous assessment. The evaluation of cytokines was further repeated by ELISA (data not shown in the manuscript due to redundancy). Moreover, in the current version of the manuscript we verified the involvement of TAOK1, -2 and -3 in IFN- β and IP-10 ELISA experiments with additional assays based on other technologies (qPCR, reporter cell assays, mass spectrometry based proteomics). We therefore confirm validity of these experiments in sufficient independent and unrelated experiments.

** Fig 7e: I don't think G-CSF is induced at all here. They should use statistics to evaluate which cytokines are actually induced and to be considered valid.*

This is correct and well spotted. Though there was a trend, we did not observe statistically significant induction of G-CSF in wt cells after SFV infection and therefore removed this dataset.

Other comments:

4) There is a confusion between "nucleic acid binding" and "proteins that associate with nucleic acids" (e.g. line 2-4 of abstract, 1st paragraph of page 6). For many readers, "binding" implies a direct interaction. The current mass spec approach identifies association or partners, but not direct binding. Few candidates are evaluated by IP (Fig S1b), but again this does not test direct interaction. Direct binding is only demonstrated for dTao (Fig 5b), which is very nice. They should carefully reformulate the text to avoid the confusion between binding and association.

This is a very valid point. We apologize for having not been correct here. We took care to rephrase the described associations to not claim direct binding for any protein other than for dTao, for which we show direct binding in Fig 5b.

5) Fig S1a, S3, S6, 2: Heatmaps show 3 columns per condition but this is not explained in the legend. Also, it could help to use different text colors for the labels of test NAs vs control NAs.

We thank the reviewer for highlight this and now colored all baits, which were used as controls, in grey. Moreover, we better describe the number of replicates per NA bait in the figure legend.

6) *Fig 1b: CDKN2AIP is mentioned in the text but I do not see it in the figure.*

The protein was present but due to the numbers of proteins shown not easily visible. We adjusted the stroke color of confirmed NA binders and of candidate proteins to achieve a better visibility.

7) *Fig 1c: I don't understand the categories shown.*

We thank the reviewer for this remark and renamed the “Human AP-MS dataset” category into “Identified human NA binders (this study)” to clarify that we compare our data with the data from either the “IAV functional screen” (Tripathi et al., 2015) or the “RNA BPome” data (Garcia-Moreno et al., 2019), both referred to as “Comparing dataset”. In addition, category colors were slightly modified to further improve visibility.

8) *Page 7: They highlight MATR3, SFPQ, PSPC1, NONO and RBM14 in association with the RNA NAs. These proteins are all members of the paraspeckle complex (e.g. PMID 28977530, 29289458) that contains RNA. However, they refer instead to the HDP-RNP, proposed in one study, which was defined by the combined associated of paraspeckle proteins with proteins involved in the DNA damage response (e.g. DNA-PK) and DNA sensing (cGAS, STING). As currently stated, they detect paraspeckles proteins co-associated with RNA NAs, but this is not an evidence for the hypothetical HDP-RNP. Is there a significant association of DNA-related HDP-RNP proteins found with the paraspeckles proteins, for given NAs? In other words, are they detecting paraspeckles, or HDP-RNP?*

We indeed found DNA-PK in RNA-NAs in certain RNA precipitates. However, we agree with reviewer 1 that our statements should be toned down in order not to overstate our findings and to be cautious in our data interpretation. In the respective paragraph we therefore now refer to enrichment of paraspeckles rather than the more specific HNP-RNP complex with its associated functionality. We feel that this finding is equally important since paraspeckle-associated proteins are linked to a variety of functions including regulation of antiviral immunity.

9) *Figure S3: It is surprising that no protein appears significantly associated with the DNA ligands. There are proteins listed in Table S4/S5 that associated with dsDNA. Why exclude the DNA-binding proteins from Figure S3? Little is known about DNA sensing in fly, so it would be interesting to read their opinion on this in the discussion as well, based on their results.*

The figure in question is based on the AP-MS data in flies and was meant to show that we can identify NA binding proteins with antiviral properties. As reviewer 1 pointed out – very little is known regarding binding of fly proteins to DNA and the consequences of these interactions for antiviral immunity. Therefore none of these interactions are shown in this particular figure.

However, we identified DNA interacting proteins in flies, these data is shown in Supplementary Figure S6 as well as Supplementary table S4 and S5 of the current version of the manuscript. Since we did not study the involvement of the identified DNA binding proteins on replication of a DNA virus in the fly (reasons given as reply A to reviewer 3), we would like to refrain from discussions on DNA binding proteins in Flies. We feel that this would require an additional study focusing exclusively on fly DNA binding proteins and linked to functional experiments using a fly DNA virus, which would be indeed very interesting. Our study provides all necessary data that is required for such a study, but we believe that speculations on functionality of these proteins would be too hypothetical at this stage.

10) Fig 3b, age 12: In another recent study, UHRF1 was shown to suppress the antiviral IFN response (PMID 33616624). The conflicting results should be mentioned and perhaps discussed.

This is a very good point and a very interesting paper. In contrast to the mentioned publication we cannot observe an antiviral effect of UHRF1 in our screening system. The most likely explanation is the divergence of experimental setups used to test the functionality of this protein. It could, for instance, be that the comparably short time of gene deletion in our system does not lead to the reported upregulation of the innate immune response seen in knockout mice and that UHRF1 also has direct effects on DNA viruses. Notably, a large body of evidence points towards direct effects of DNA methylating enzymes on DNA viruses. However, such direct effects may be masked in the knockout mice due to the simultaneous effects on endogenous DNA leading to upregulation of type-I interferons. We now reference the above mentioned paper and indicate that the function of this protein is clearly interesting in the context of antiviral Immunity and would need to be further studied.

11) Fig 3b, 4b: p-val circles on red cells are not visible in print. Please use a different color scheme.

The color of p-value circles was adjusted to improve visibility with any shades of green or red in the underlying heatmap.

12) They identify DDX41 as a clear partner of RNA ligands, but not of DNA ligands. Considering the controversy in the literature on the role of DDX41 in DNA sensing, do they foresee a role of RNAs? Can they discuss this?

We agree with reviewer 1 that the literature on DDX41 is controversial. This could, at least in part, be due to the different infection models used, which would be in line with our functional data: Lack of DDX41 increases influenza A virus replication in non-differentiated THP-I cells, thus possibly expanding the role of DDX41 to RNA viruses. However, we can also see a clear effect of DDX41 in HSV-1 infection, although the activity seems limited to differentiated THP-I cells. These unbiased results point towards specific effects of DDX41 on the cell differentiation state and based on the reviewer's comment, we touch on this in the manuscript.

Reviewer #2 (Remarks to the Author):

The authors report a very nice method of unbiased screening for conserved antiviral proteins using mass spectrometry in three species: fly, mouse, and human to detect proteins binding to nucleic acids common to a diverse set of viruses. The overlapping candidates were then functionally probed for their ability to modulate viral replication by an in-vitro CRISPR-cell based screen and in vivo knockdown of targets in flies. One of the best candidates, Tao kinase was shown to have specific binding to Poly I:C. The effect of Tao kinases was also tested with some putative inhibitors of TaoK, which enhanced viral replication. TAOK's contribution to signaling was narrowed down to IRF3 but not NF-kB mediated signaling. Several new targets for investigation of anti-viral proteins have been described and warrants further follow up or may provide corroborating evidence for other current studies. There are some important changes/ additions needed to enhance the conclusions of the report.

Major critiques:

1. The role of TAOK2 is convincingly shown, but to complete the mechanistic connection to known antiviral pathways, I would suggest the authors pulldown TAOK2 and perform Mass Spectrometry to identify binding partners (IP-MS). This could be performed across species or in a single species with confirmation by Western Blot. Given the author's expertise in MS assays and cell manipulations, this should be an achievable experiment that will provide convincing evidence of the role of TAOK2.

We thank reviewer 2 for this comment, which prompted us to invest extensive effort to further study the potential mechanism of TAOK2 activity. To this aim we performed AP-MS experiments using specific antibodies to precipitate endogenous TAOK2 in wt and TAOK2 KO cells. However, the endogenous antibodies used did not allow sufficient enrichment to faithfully identify binding partners (data not shown). We therefore expressed tagged versions of rat TAOK2 (we were not able to clone human TAOK2), a kinase dead mutant TAOK2 (D151A) and a TAOK2 (R700C) mutant, which was identified in clinical studies to be associated with generalized verrucosis, a human papilloma virus induced disease. To stimulate potential TAOK2 activity and to induce potential complex formation we performed AP-MS experiments in absence or presence of poly-I:C transfection. The obtained data is remarkable: TAOK2 binds to proteins linked to its published functions in DNA damage repair and cell death regulation. However, the main interactor of TAOK2 was TRIM4, one of the key ubiquitin ligases that adds K68-linked ubiquitin to stimulate RIG-I activity. Notably, while TAOK2 associates to TRIM4 independently of its kinase activity, this association is critically requiring R700, which is mutated in patients with generalized verrucosis. We are very intrigued by these experiments since it may provide a mechanistic explanation regarding the TAOK2 involvement in type-I interferon induction. These data is now provided as new Figure 8 and Supplementary Figure S9 and Supplementary Table S12) While these data may provide a mechanistic link to innate immune signalling it is not yet clear how TAOK2 selectively contributes to type-I interferon rather than proinflammatory cytokine expression. However, we feel that the answer

to this very important question would exceed the scope of this manuscript, which is rather focused on the characterization of the repertoire of pathogen-derived nucleic acid sensing. The newly provided data will, however, spark further mechanistic studies linking TAOK2 to antiviral processes.

2. The data in Figure 4 are currently organized alphabetically, but I would suggest they be organized based on the nucleic acid type they bound in the initial screen. This would allow for a direct comparison of how they impacted viral replication. Ex: proteins binding to DNA vs. the dsDNA virus HSV1.

We very much appreciate the reviewer's comment and accordingly annotated the NA binding specificity of each candidate protein according to the three NA categories DNA, RNA and 2'5'OA (Figure 3b and Figure 4b). In addition, we provided an overview figure with the most relevant functional interactions and detailed NA binding characteristics as Figure 8b (Figure 9b in the current version of the manuscript). We also tried a variety of unbiased clustering approaches and combination of the datasets. However, the issue here was, that the 11 nucleic acids used for AP-MS and the selection of „only“ 100 candidates does not lead to an intuitive clustering. Some candidates are part of several categories (e.g. DNA binding and poly-I:C binding), which does not lead to the improved clarity that one would hope for. A logical clustering was therefore not possible and it was rather difficult to find the individually discussed proteins when ordered in a non-alphabetical order. We ask reviewer 2 for his understanding that we would rather present the data in alphabetical order, but with additional annotation of the NA type.

3. Statistical comments: For figures 3b and 4b, I suggest changing -log₁₀ p-value from 10 to 0.01 etc. It was confusing to interpret without reading a few times.

a. For Figure 6, and 7, please include p-values between matched conditions.

We thank the reviewer for his/her comments regarding the statistical analyses. Accordingly, we applied a two-way ANOVA with Šídák's multiple comparisons testing for Figure 7 to identify statistically significant induction of cytokines upon SFV infection and TAOK2 knockout cells. To determine statistically significant changes in case of the live-cell imaging data presented in Figure 6, we used a repeated measurements two-way ANOVA analysis with Šídák's multiple comparisons testing, which allows statistical comparison of treatment versus control conditions. In general, lines and error bars in these figures represent the mean +/- s.d. and statistical annotations were generalized as follows: ns, p-value > 0.05; *, p-value < 0.05; **, p-value < 0.01; ***, p-value < 0.001; ****, p-value < 0.0001. This is now also described in the figure legends.

Regarding the suggestion to transform -log₁₀ p-values back to a linear scale: to our knowledge this would be a rather uncommon adjustment since a -log₁₀ p-value of 10 corresponds to a p-value of 0.0000000001 (10⁻¹⁰). To compress the range of p-values and optimize them for visualization we chose to display the data in a -log₁₀ transformed manner, which is commonly

used for such low p-values. However, we would be happy to change this, if wanted form editorial site too. Otherwise we would rather prefer the current display, and ask for the reviewer's consent.

Minor comments:

1. *The 2nd sentence and second to last sentence of the first introductory paragraph and vague and specific cytokines and cellular machineries should be named or re-worded. Additionally "specific marks" line 5 paragraph 2 of introduction should be more clear.*

This is a well taken point, we changed the text to be more precise.

2. *Italicize gene names where appropriate (human genes).*

Gene names are now shown in *italics*

3. *Spell out AP-MS the first time the abbreviation is used.*

Done

4. *Pg 5 line 4, please provide a reference for the NA binding properties of IFIT1, RIG-I and DHX9.*

Fixed

5. *SMARCA5 had some of the strongest viral replication present when knocked out (Fig. 3B). Why was it not followed up on for investigation?*

SMARCA5 is indeed an interesting candidate. We were aware of another laboratory actively working already on SMARCA proteins and we did not want to get in conflict with their work. Since the data is on the NA interactions and the screen are presented and the protein is mentioned in the text, we hope that other scientists will pick up this candidate for further studies.

6. *Page 20: The evidence provided suggests, but does not "indicate" TAOK evolved in parallel to TLR and cGAS- more advanced phylogeny and genetic studies would be needed for this conclusion. Also, what is the point of the next sentence after this. It appears as non-sequeter and does not summarize the preceding paragraph nor transition to the next. I suggest moving or removing.*

We removed the sentence, as suggested.

7. *Though not required for this study, knocking out TAOK2 or other targets in mice and performing comprehensive immune analysis would be a fruitful future direction.*

We agree that this would be very informative. However, to our knowledge the TAOK2 depletion in mice is lethal (as is Tao depletion in flies), suggesting that a straight knockout would not be very easy to perform. We currently envision to introduce a TAOK2(R700C) mutation in mice, which should be viable given that this mutation was identified in patients with generalized verrucosis.

Signed,
Jeff SoRelle, MD

Reviewer #3 (Remarks to the Author):

In the manuscript by Pennemann et al., the authors describe a screen for host nucleic acid sensors using human, mouse and fly model systems. The immense amount of data generated through this screen is analyzed well, and experiments to validate the hits uncovered in the screen are clearly performed using the relevant model systems. The authors' tour de force in the screen and validation is highly commendable. Following the experimental validation of the protein candidates, the authors focus on the TAO kinases for functional assays. This section needs further development to justify the authors conclusions, e.g., a statement in the abstract regarding TAOK2 and regulation of IRF3 versus NFkB. Also, this section mostly focuses on the mammalian TAOs even though dTao was quantitatively analyzed for dsRNA binding and activation. Below are suggestions to the authors that would improve their manuscript.

MAJOR:

A) Figure 4 should include a viral pathogen with a dsDNA genome, similar to the validation experiment with HSV-1 and the mammalian CRISPR screen.

We very much appreciate the positive comment of reviewer 3. We chose to focus on RNA viruses because metagenomic analysis revealed that the vast majority of viruses associated with *Drosophila* have an RNA genome. In fact, only one DNA virus has been found to be associated with *Drosophila melanogaster* so far (Webster et al., PLoS Biol. 2015 Jul 14;13(7):e1002210). We want to stress that the fly screen already involved five different viruses and two time points for each, which represented a significant time investment (two years to set up the screen and conduct it). We made the strategic decision to use only RNA viruses based on the observation that (i) viruses isolated from natural populations of flies are overwhelmingly RNA viruses; (ii) several innate immunity receptors detecting DNA virus infections sense RNA and also show phenotypes with RNA viruses (TLR3; RNaseL-RIG-I; Dicer-2 in flies); (iii) the top hits for the dsISD bait in *Drosophila* samples were known host DNA binding proteins and did not include promising candidates. Repeating this work with a DNA virus would take another six months of work including viral production, flies stocks, crosses and RT-qPCR. We ask for her/his understanding that a repetition of this screen in living flies cannot be conducted for resource reasons. To acknowledge the criticism of the referee we have added one sentence on page 13 to justify the use of RNA virus pathogens.

B) Figure 5/Page 20 – the authors state that TAOs and dTao bind dsRNA with high affinity. Without a proper control for the experiments in Figure 5A-C, the conclusion is not fully justified. The authors should include known mammalian and fly dsRNA interactors in these experiments.

We are a bit surprised by this comment: These are validation experiments, which confirm the data obtained by the AP-MS screen. Therefore these experiments are not stand alone figures and are correctly controlled:

Figure 5a: The confirmation of TAO 1, -2, -3 binding to poly-I:C (Figure 5a) is clearly controlled since TAO kinases do not associate to poly-C, as shown in this western blot. The unrelated control protein β -actin, did not associate with poly-I:C. The specificity of the poly-I:C precipitation approach is best evident from the AP-MS results, which show clearly higher enrichment of DIRCR2, RIG-I, LCP3 (DHX58) and PKR (EIF2AK2) in poly-I:C vs poly-C precipitates (Supplementary Fig. 1a). In addition, we show selective binding of the TAO kinase unrelated protein SMARCA5 to poly-I:C (Supplementary Fig. 1b). We therefore feel that the use of a yet additional known dsRNA binding protein in the western blotting experiments would not add additional information.

Figure 5b: Microscale thermophoresis is a commonly accepted technology to test the affinity of two entities. It is in this sense similar to surface plasmon resonance (SPR) measurements. In such experiments it is not common to provide “control experiments” with unrelated proteins as positive controls. We used denatured protein as negative control, which clearly lacks any binding capacity, indicating that the obtained values are not due to experimental issues. Using a dsRNA binding positive control would require generating and purifying a recombinant dsRNA binding protein to measure the affinity of this particular protein for control purposes only. We feel that this analysis for the affinity of a known control protein does not add additional information on binding of TAO kinases to dsRNA. In light of the data that went into this manuscript already, we ask reviewer 2 for his/her understanding that we did not perform these experiments.

Figure 5c: We are further surprised by the comment on Figure 5c, which is an ATPase assay often used for kinase activation assays. We can show that addition of poly-I:C to recombinant dTao increases its ATP consumption. This finding is in our opinion striking. We are not sure how using an additional unrelated dsRNA binding protein with kinase activity would increase our knowledge on Tao. Regardless if ATPase activity may increase similarly or differently when comparing two unrelated proteins we would not know what that means in functional terms. In this figure we show, however, that the same protein (Tao) changes ATPase activity upon dsRNA addition.

C) Even though the authors provide quantitative data showing that dTao interacts with and is activated by dsRNA, no further functional validation is performed in the fly model. This is

surprising considering the number of reagents available to evaluate dTao in a fly model. These experiments are needed to support the role of dTao in the fly.

We thank reviewer 3 for this comment. We performed additional experiments in flies, which aimed at showing an antiviral effect of Tao in flies. Unfortunately, Tao turned out to be an essential gene, required for survival of *Drosophila melanogaster*. When silenced in adults, we observed that flies succumb even when injected with the buffer, which prevents a rigorous analysis of survival to viral infections. In the revised version of the manuscript we show this data in Supplementary Figure 7a and describe the phenotype in the results section. However, we monitored the expression of the marker genes *srg-1* and *diedel* two days after infection with DCV. As expected, infection of control flies induces expression of these genes in flies. In contrast, Tao depleted flies do not show induction in response to DCV infection underlining that Tao is also important for the antiviral immune response in flies. We also include these new data as Supplementary Figure 7b, c and discuss it in the result section. We thank reviewer 3 for this comment since it is an excellent example that the cross-species nucleic acid affinity screen is able to identify relevant proteins in mammalian and flies and further underlines the importance of TAO kinases in antiviral immunity.

D) RAF265 and Compound 43 function on the host responses, both in the context of control and TAOK2 KO, should be examined, as described in Koo et al.

We thank Reviewer 3 for this comment, which was also a major point of reviewer 1. In order to test the specificity of RAF265 we treated wt and TAOK2 knockout A549-IFIT1-GFP cells with the inhibitor and monitored IFIT1-GFP expression in response to SFV infection using time-lapse microscopy. As expected, compared to wt cells, TAOK2 deficiency led to reduced expression of IFIT1-GFP. However, while RAF265 was active to suppress IFIT1-GFP in wt cells, the inhibitor only showed minor activity in TAOK2 deficient A549-IFIT1-GFP cells. These data clearly shows that RAF265's inhibition of IFIT1-GFP expression is operating through inhibition of TAOK2. The new data is now provided as Supplementary Figure 7f. Since we had only used Compound 43 in a single experiment we decided to remove this data from the current manuscript.

E) Page 17 – the authors conclude that TAOK2 affects IRF3 but not NFkB (also stated in the abstract). However, the data do not fully support this conclusion. In the same paragraph they state that MCP-1 is induced by NFkB, and the data show that MCP-1 secretion is indeed affected by TAOK2. Nevertheless, the authors should perform assays to directly test IRF3 and NFkB activation during SFV infection and TAOK2 KO to properly support their conclusions.

We agree with reviewer 3 that our previous comments on IRF3 and NF-kB activation were overstated since we did not show directly that these transcription factors are or are not activated. The involvement of these transcription factors was rather deduced from the very specific protein expression signature that is elicited in TAOK knockout cells. In the current version of the manuscript we toned down our conclusions and only relate to an involvement of TAO kinases in cytokine induction and antiviral signaling. This statement is backed up with numerous figures in the current version of the manuscript. Our new data shows TAOK2

association to TRIM4, indicating a more complex mechanistic engagement of TAOK2 in antiviral signaling. We will further validate the TAOK2-TRIM4 interaction in future studies and in particular research the relationship to transcription factor activation and ask reviewer 3 for his/her understanding that it was not possible for us to establish the phospho-western blotting assays in our laboratory during this already quite intense revisions.

MINOR:

A) In the abstract, the authors state 182 candidates, 98 restriction factors, and 16 proteins, but these numbers do not agree with what is presented in Figure 8. Further, in Figure 8, are the 28 proteins shown in panel B the 28 at the intersection of the “Candidates” Venn diagram? Which are the 14 that validated by functional screening?

We agree with Reviewer 3 that these numbers without any additional clarification were more confusing than helpful. We have thus optimized the legend of new Figure 9 (old Figure 8) to better explain these numbers and further combined the number of anti- and proviral proteins, which were mentioned in the abstract, into “proteins with an impact on viral growth”.

B) The introduction section should briefly describe the results from the functional assays on TAOK shown in Figures 5-7.

We now briefly describe the findings on TAO kinases in the introduction.

C) Page 13 – dTao should be discussed in the fly functional screen section.

We had mentioned dTao function in the fly screening result. In the revised manuscript we provide additional data on challenge experiments in dTao knockdown flies. We opted to add these data to the functional evaluation of human TAOK since we feel that it fits better to this section and shows that the function of TAO kinases is conserved between species.

D) Figure 7 – statistical tests should be shown for each figure panel.

We thank the reviewer for his/her comments regarding the statistical analyses. Accordingly, we applied a two-way ANOVA with Šídák's multiple comparisons testing for Figure 7 to identify statistically significant induction of cytokines upon SFV infection and TAOK2 knock-out. In general, lines and error bars in these figures represent the mean +/- s.d. and statistical annotations were generalized as follows: ns, p-value > 0.05; *, p-value < 0.05; **, p-value < 0.01; ***, p-value < 0.001; ****, p-value < 0.0001. We added this information to figure legends.

E) Discussion – the authors describe the importance of cGAS at the beginning of this section, but cGAS was not found in their screen. Can they speculate about why cGAS was absent from the initial screening results?

This is correct, this is very well spotted, we did not identify cGAS in the screening approach. There are two reasons why this is an expected result:

1. Considering the average binding affinity of nucleic acid binding proteins, the affinity of cGAS to DNA is actually relatively low ($KD > 100$ nM), in particular when short DNAs are used, such as in our case. Due to this low affinity, recombinant cGAS does, for instance, not even co-migrate with DNA on gel filtration columns in highly controlled conditions. It may be that special features of DNA, such as present on Y-Form DNA, would boost the affinity and thus allow cGAS enrichment. However, we are not aware of any study that identified cGAS binding to DNA by unbiased AP-MS experiments.
2. The majority of cGAS is nuclear and the lysis conditions required for the NA-AP-MS approach used in our study would only release minute amounts of cGAS into cell lysates. The relatively high affinity for cGAS to chromatinized DNA (~10-times higher than affinity to DNA), would, in addition, lead to precipitation of cGAS into insoluble cell fractions – i.e. the cell pellet.

REVIEWERS' COMMENTS

Reviewer #1 (Remarks to the Author):

The study has greatly improved. Congratulations to the authors for a very interesting resource and intriguing new findings. I have two remaining comments:

1) The new data identifying TRIM4 as a partner of the TAOK2 complex is interesting. However they have not tested it functionally and this leaves a significant gap in the study. The speculation that TAOK2 might implicate RIG-I through TRIM4 is confusing (l.508-510; l.652-653). They clearly show TAOK2 is not implicated in the IFN response to a RIG-I ligand (in response to one of my previous comments) and is specific to poly(I:C). One study previously showed that the IFN response to SFV is entirely dependent on MDA5, and there is only a minor contribution of RIG-I, in the conditions tested (PMID 20478537). Isn't it simpler to think that TRIM4 could regulate MDA5 signaling directly? Since they have not yet tested the role of TRIM4 or MDA5 in their system, it might be worth expanding or clarify this.

2) l.366-368: Please indicate which figure panel you are refereeing to for this statement.

Reviewer #2 (Remarks to the Author):

The extensive work performed by the authors not only improved the quality of the mechanistic portion of their manuscript, but opened up new avenues of future research. The results of consistent and strong binding of TRIM4 to TAOK2 was impressive and exciting, mechanistically linked TAOK2 to anti-viral immunity. Their decision to even examine kinase dead and human variants in their interactions went above and beyond what was asked for, but provide amino acid level resolution of molecular mechanism.

All of my other comments are satisfied, and I appreciated the now enhanced figures 3 and 4 which outline which NAs are involved in each case.

Best of luck in your future endeavors.

Reviewer #3 (Remarks to the Author):

The authors have addressed the reviewers' previous concerns through the addition of new data or clarifications in the text. This is a comprehensive and well-organised study for which the authors should be congratulated!

Reply to the Reviewers' comments

We want to thank all reviewers for their positive and very constructive comments throughout this revision. Below you find the response to the remaining points raised by Reviewer #1.

With best regards,
Andreas Pichlmair and the authors of this manuscript

REVIEWERS' COMMENTS

Reviewer #1 (Remarks to the Author):

The study has greatly improved. Congratulations to the authors for a very interesting resource and intriguing new findings. I have two remaining comments:

1) The new data identifying TRIM4 as a partner of the TAOK2 complex is interesting. However they have not tested it functionally and this leaves a significant gap in the study. The speculation that TAOK2 might implicate RIG-I through TRIM4 is confusing (l.508-510; l.652-653). They clearly show TAOK2 is not implicated in the IFN response to a RIG-I ligand (in response to one of my previous comments) and is specific to poly(I:C). One study previously showed that the IFN response to SFV is entirely dependent on MDA5, and there is only a minor contribution of RIG-I, in the conditions tested (PMID 20478537). Isn't it simpler to think that TRIM4 could regulate MDA5 signaling directly? Since they have not yet tested the role of TRIM4 or MDA5 in their system, it might be worth expanding or clarify this.

We thank Reviewer #1 for the very thoughtful remark regarding a possible regulation of MDA5 by TRIM4. When describing the link between RIG-I and TRIM4 we cited the original research paper reporting the K68-dependent ubiquitination of RIG-I by TRIM4. In light of the phenotype observed upon TAOK2 KO and SFV infection, we agree that an involvement of MDA5 might be plausible. In case of poly(I:C) stimulation, we already discussed a potential regulation of MDA5 by TRIM4 (lines 639-613). In addition, we now discuss the potential involvement of MDA5/TRIM4 in SFV infection and TAOK2 KO (new lines XXX-XXX).

“...The consequence of this interaction remains to be further evaluated, but it may indicate yet unexplored functional relationships that could be relevant for antiviral immunity. Poly(I:C), which is binding and activating TAOK2, is sensed by the PRRs MDA5 and TLR3. TRIM4, which has originally been found to activate RIG-I by K68-dependent ubiquitination, may also regulate activity of MDA5 or TLR3. An involvement of TAOK2 and TRIM4 in MDA5 dependent responses is further supported by the lack of IFN production in TAOK2 KO cells after infection with SFV, which is a MDA5 activating

virus. Very little is known about the regulation of E3 ligases involved in innate immunity. However, one could speculate that TAOK2 regulates the activity of TRIM4 that in turn activate RIG-I, MDA5 or Toll like receptor signalling through its E3 ligase activity.”

2) 1.366-368: Please indicate which figure panel you are refereeing to for this statement.

We added a reference to Figure 3b and Figure 4b for these statements.

Reviewer #2 (Remarks to the Author):

The extensive work performed by the authors not only improved the quality of the mechanistic portion of their manuscript, but opened up new avenues of future research. The results of consistent and strong binding of TRIM4 to TAOK2 was impressive and exciting, mechanistically linked TAOK2 to anti-viral immunity. Their decision to even examine kinase dead and human variants in their interactions went above and beyond what was asked for, but provide amino acid level resolution of molecular mechanism.

All of my other comments are satisfied, and I appreciated the now enhanced figures 3 and 4 which outline which NAs are involved in each case.

Best of luck in your future endeavors.

Reviewer #3 (Remarks to the Author):

The authors have addressed the reviewers' previous concerns through the addition of new data or clarifications in the text. This is a comprehensive and well-organised study for which the authors should be congratulated!